# Calcium-sensing receptor-mediated NLRP3 inflammasome response to calciprotein particles drives inflammation in rheumatoid arthritis

Elisabeth Jäger [1,9], Supriya Murthy [1,9], Caroline Schmidt [1], Magdalena Hahn [1], Sarah Strobel[1], Anna Peters [2], Claudia Stäubert [2], Pelin Sungur [3], Tom Venus[4], Mandy Geisler[5], Veselina Radusheva[1], Stefanie Raps[1], Kathrin Rothe[1], Roger Scholz[6], Sebastian Jung[1], Sylke Wagner[1], Matthias Pierer[1], Olga Seifert[1], Wenhan Chang [7], Irina Estrela-Lopis[4], Nora Raulien[1], Knut Krohn [8], Norbert Sträter[5], Stephanie Hoeppener [3], Torsten Schöneberg[2], Manuela Rossol [1,10✉] & Ulf Wagner [1,10✉]

Increased extracellular Ca$^{2+}$ concentrations ([Ca$^{2+}$]$_{ex}$) trigger activation of the NLRP3 inflammasome in monocytes through calcium-sensing receptor (CaSR). To prevent extra-osseous calcification in vivo, the serum protein fetuin-A stabilizes calcium and phosphate into 70-100 nm-sized colloidal calciprotein particles (CPPs). Here we show that monocytes engulf CPPs via macropinocytosis, and this process is strictly dependent on CaSR signaling triggered by increases in [Ca$^{2+}$]$_{ex}$. Enhanced macropinocytosis of CPPs results in increased lysosomal activity, NLRP3 inflammasome activation, and IL-1β release. Monocytes in the context of rheumatoid arthritis (RA) exhibit increased CPP uptake and IL-1β release in response to CaSR signaling. CaSR expression in these monocytes and local [Ca$^{2+}$] in afflicted joints are increased, probably contributing to this enhanced response. We propose that CaSR-mediated NLRP3 inflammasome activation contributes to inflammatory arthritis and systemic inflammation not only in RA, but possibly also in other inflammatory conditions. Inhibition of CaSR-mediated CPP uptake might be a therapeutic approach to treating RA.

[1] Department of Internal Medicine, Division of Rheumatology, Leipzig University, Liebigstraße 19, 04103 Leipzig, Germany. [2] Rudolf Schönheimer Institute of Biochemistry, Faculty of Medicine, Leipzig University, Johannisallee 30, 04103 Leipzig, Germany. [3] Jena Center for Soft Matter (JCSM), Friedrich-Schiller-University Jena, Humboldtstraße 10, 07743 Jena, Germany. [4] Institute for Medical Physics and Biophysics, Leipzig University, Härtelstraße 16-18, 04107 Leipzig, Germany. [5] Institute of Bioanalytical Chemistry, Center for Biotechnology and Biomedicine, Leipzig University, Deutscher Platz 5, 04103 Leipzig, Germany. [6] Department of Orthopaedic, Trauma and Plastic Surgery, Leipzig University, Liebigstraße 20, Leipzig, Germany. [7] UCSF Department of Veterans Affairs Medical Center, San Francisco, CA, USA. [8] DNA Core Unit Leipzig, Liebigstraße 19, 04103 Leipzig, Germany. [9]These authors contributed equally: Elisabeth Jäger, Supriya Murthy. [10]These authors jointly supervised this work: Manuela Rossol, Ulf Wagner. ✉email: manuela.rossol@medizin.uni-leipzig.de; ulf.wagner@medizin.uni-leipzig.de

The calcium-sensing receptor (CaSR) has been shown to trigger NLRP3 inflammasome activation both in vivo and in vitro[1,2] and has been described to mediate inflammatory effects in human cryopyrin-associated periodic syndromes (CAPS)[1] and in the carrageenan-induced foot pad swelling mouse model[2]. Increased $[Ca^{2+}]_{ex}$ also triggers constitutive macro-pinocytosis in monocytes and macrophages via G protein-mediated CaSR signaling[3]. The receptor is expressed in numerous cell types, such as monocytes, macrophages, smooth muscle and endothelial cells, and contributes to inflammatory responses in allergic asthma[4] and inflammatory lung disease[5], after myocardial infarction[6] and in obesity[7].

Short-lived increases in $[Ca^{2+}]_{ex}$ occur in the interstitial fluid around activated[8–10] or dying cells[2]. More longstanding increases of $[Ca^{2+}]_{ex}$ have been reported at sites of chronic infections[11–13] and dialysis-related peritonitis[14]. Homeostasis of $[Ca^{2+}]_{ex}$ is intricately linked with the anion phosphate $[P_i]$, the clinical relevance of which is most apparent in end-stage chronic kidney disease (CKD). In those patients, hyperphosphatemia is associated with excess mortality, systemic inflammation and vascular calcification[15], especially in cases of severe hyperphosphatemia which is defined as serum levels higher than 4.54 mM[16].

Coronary artery calcifications, which indicate perturbations of $[Ca^{2+}]_{ex}$ and $[P_i]$ homeostasis, occur in CKD[17,18], but also in coronary artery disease and other vascular degenerative diseases[19,20]. Failure to prevent ectopic mineralization due to $[Ca^{2+}]_{ex}$ and $[P_i]$ overload is common in tissues with localized inflammation and cell necrosis[21,22], and occurs also in aorta and carotid and coronary arteries in the autoimmune disease rheumatoid arthritis (RA)[23]. Importantly, activation of the NLRP3 inflammasome has been linked to vascular damage and excess mortality in atherosclerosis[24] and myocardial infarction[25]. Clinical evidence for an aggravating effect of inflammasome activation in atherosclerosis comes from the CANTOS study, which showed the benefit of IL-1β blockade in patients with previous myocardial infarction[26].

In RA, increased CaSR expression has been associated with severe coronary artery calcification[27], but the role of the receptor in arthritis has not been investigated. In this disease, the invading pannus forms an osteo-immunological interphase with the bone matrix at the sites of erosive lesions. At erosion sites beneath osteoclasts, extremely high $[Ca^{2+}]_{ex}$ values of 40 mM have been measured[28]. Indirect evidence suggests that besides $Ca^{2+}$, $P_i$, and the $Ca^{2+}$-binding crystallization inhibitor fetuin-A are also liberated from the extracellular matrix during osteoclastic bone resorption[29–31]. Fetuin-A is known to stabilize calcium phosphate crystal precursors during bone mineralization as colloids and is one of the most abundant non-collagenous proteins in bone[32].

This study investigates the mechanism of $[Ca^{2+}]_{ex}$-induced IL-1β release in RA, and the function of $[P_i]_{ex}$ in $[Ca^{2+}]_{ex}$-mediated inflammasome activation. We find that increased $[Ca^{2+}]_{ex}$ in the presence of increased $[P_i]_{ex}$ and fetuin-A leads to the formation of fetuin-A-based calciprotein particles (CPPs), and simultaneously induces CaSR signaling, which triggers CPP uptake and subsequently NLRP3 inflammasome activation. This CaSR-mediated process and the resulting IL-1β release are enhanced in RA. Allosteric enhancement of CaSR signaling in vivo leads to aggravation of arthritis, which emphasizes the pivotal role of the receptor in this disease.

## Results

### Increased $[Ca^{2+}]_{ex}$ leads to calciprotein particle formation.

To explore CaSR-triggered intracellular events leading to NLRP3-dependent IL-1β production in myeloid cells, a CaSR-deficient monocytic THP-1 cell line was established using CRISPR-Cas9

technology (CRISPR–Cas9-CaSR). The $[Ca^{2+}]_{ex}$-induced IL-1β response was found to be significantly diminished in CaSR-deficient THP-1 cells, while their response to adenosine triphosphate (ATP) or monosodium urate (MSU) crystals was unaffected (Fig. 1a). This CaSR effect was confirmed by experiments with peripheral blood monocytes from mice with a myeloid-specific CaSR ablation (B6.129P2-Lyz2$^{tm1(cre)Ifo7}$x-CaSR$^{\Delta flox/\Delta flox}$), which were found to respond with significantly decreased IL-1β secretion upon stimulation with $[Ca^{2+}]_{ex}$ in the presence of 5.6 mM $[P_i]_{ex}$ (Fig. 1b), while their ATP response was unaltered (Supplementary Fig. 1a). $[Ca^{2+}]_{ex}$-induced IL-1β responses were dependent on the NLRP3 inflammasome both in THP-1 cells and in peripheral blood monocytes, as demonstrated by experiments with a NLRP3 knock-down THP-1 cell line and with peripheral blood monocytes treated with the NLRP3-specific inhibitor MCC950 (Supplementary Fig. 1b, c).

As shown previously[2], $[Ca^{2+}]_{ex}$-induced NLRP3 inflammasome activation does not occur when monocytes are primed with lipopolysaccharide (LPS) and stimulated only briefly with increased $[Ca^{2+}]_{ex}$, which is in contrast to the ATP response. Maximum IL-1β release occurs when LPS is added simultaneously with increased $[Ca^{2+}]_{ex}$, but requires considerably longer stimulation compared to ATP (Supplementary Fig. 1d). Accordingly, in all subsequent experiments monocytes were stimulated with $[Ca^{2+}]_{ex}$ and LPS simultaneously.

The influence of $[P_i]_{ex}$ on $[Ca^{2+}]_{ex}$-induced IL-1β release in monocytes was investigated using customized RPMI1640 media with $[P_i]$ ranging from 1 to 5.6 mM, whereby $[Ca^{2+}]$ was added as calcium chloride. Due to protein binding, the final concentration of ionized calcium in the tissue culture medium is always lower than the added calcium chloride concentration, which is indicated in all figures (detailed analysis in Rossol et al., 2012[2]) unless indicated otherwise. At ≥3 mM of $[P_i]_{ex}$, increased $[Ca^{2+}]_{ex}$-triggered concentration-dependent IL-1β release (Fig. 1c) and ASC (apoptosis-associated speck-like protein containing a CARD) speck formation (Supplementary Fig. 1e) was observed. No $[Ca^{2+}]_{ex}$ effect was discernible at low $[P_i]_{ex}$. High $[P_i]_{ex}$ combined with low $[Ca^{2+}]_{ex}$ did also not induce IL-1β release (Fig. 1c).

In serum-free buffer or medium, an increase of $[Ca^{2+}]$ and $[P_i]$ beyond the solubility product immediately leads to precipitation of macroscopically visible calcium orthophosphate crystals. In vivo, calcium phosphate precipitation in body fluids is prevented by the presence of the serum protein fetuin-A and other $Ca^{2+}$-binding proteins. Fetuin-A is present in high concentrations in fetal bovine serum (FBS), and addition of FBS completely prevented the microscopically detectable calcium orthophosphate crystal formation in RPMI1640 cell culture medium upon addition of 2.5 mM $Ca^{2+}$. However, using dynamic light scattering analysis (DLS), we were able to detect nanometer-sized particles if $Ca^{2+}$ was added to RPMI1640/10% FBS. The particles were below the detection limit of light microscopy due to their size between 60 and 100 nm, but were detectable by DLS already seconds after addition of $Ca^{2+}$ to FBS-supplemented culture medium (Fig. 1d, e). The particles developed only at ≥3 mM of $[P_i]_{ex}$, and not in the absence of $[Ca^{2+}]$ (see Fig. 1d). Their size increased only marginally over 20 h of incubation (Fig. 1e). In parallel to particle formation, $[Ca^{2+}]$ decreased due to the incorporation of $Ca^{2+}$ in particles. The added $Ca^{2+}$ was completely used up by this process within 20 h (Fig. 1e).

Using high-speed centrifugation (16,000 × g) over 2 h, we sedimented these particles, thereby making the pellet macroscopically visible (Supplementary Fig. 2a). Precipitates forming in serum-free RPMI1640 cell culture medium upon addition of 2.5 mM $[Ca^{2+}]$ were clearly visible already in suspension and formed a substantial white pellet after centrifugation. The

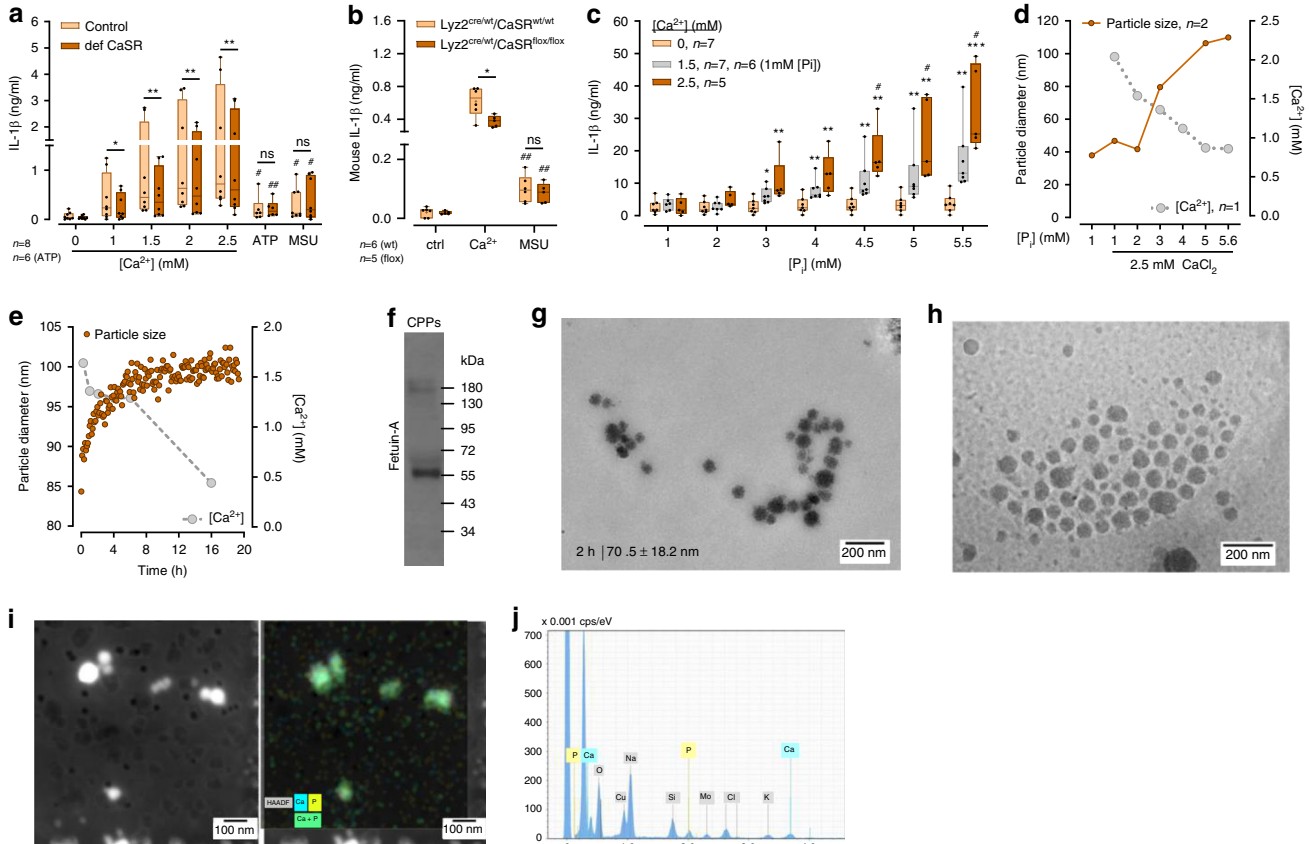

**Fig. 1 In the presence of [$P_i$] and fetuin-A, addition of [$Ca^{2+}$] triggers calciprotein particle formation. a** Differentiated CaSR-deficient (def-CaSR) or control THP-1 cells incubated with LPS and either the indicated amount of [$Ca^{2+}$], 3 mM ATP, or 100 μg/ml MSU in RPMI1640/10%FBS containing 5.6 mM [$P_i$] for 8 h (numbers of experiments as indicated). **b** Blood monocytes of myeloid CaSR-KO mice (B6.129P2-Lyz2$^{tm1(cre)Ifo7j}$xCaSR$^{\Delta flox/\Delta flox}$, $n = 9$) or control mice (B6.129P2-Lyz2$^{tm1(cre)Ifo7j}$xCaSR$^{WT/WT}$, $n = 8$) were treated for 16 h with LPS (ctrl), LPS plus 2.5 mM added [$Ca^{2+}$] or LPS plus 100 μg/ml MSU in RPMI1640/10%FBS containing 5.6 mM [$P_i$]. Blood from 2 to 3 mice was pooled, and used in three experiments. **c** Freshly isolated human blood monocytes from the indicated numbers of donors were incubated with LPS and the indicated amount of [$Ca^{2+}$] in RPMI1640/10%FBS containing the indicated [$P_i$] for 16 h. **a**–**c** IL-1β was detected in supernatants. Box-and-Whisker plots show median, 25–75th percentile, and min/max Whiskers. Wilcoxon signed-rank test **a** or two-tailed Mann–Whitney U test **b**, **c** was used. p-values are indicated as *$p < 0.05$, **$p < 0.01$, ***$p < 0.001$, ****$p < 0.0001$ (or # respectively). #Indicates level of significance for comparison to control (LPS) in **b** and for comparison between 1.5 and 2.5 mM added [$Ca^{2+}$] in **c**; *Indicates level of significance for comparison between 0 and 2.5 mM added [$Ca^{2+}$] in **c**. **d**, **e** Dynamic light scattering analysis (main peak intensity) of nanoparticles forming in RPMI1640/10%FBS. **d** Particle size (two experiments with technical triplicates) and [$Ca^{2+}$] (one experiment) in relation to [$P_i$] after addition of 2.5 mM [$Ca^{2+}$]. Data show mean. **e** Time course of particle size and [$Ca^{2+}$] after addition of 2.5 mM CaCl$_2$ to RPMI1640/10%FBS with 5.6 mM [$P_i$]. **f** Western Blot detecting fetuin-A in isolated calciprotein particles (CPPs). Shown is one representative experiment out of three. **g**, **h** Transmission electron microscopy (TEM) and Cryo-TEM imaging ($n = 2$) of particles formed spontaneously after 2 h in RPMI1640/10%FBS and 2.5 mM [$Ca^{2+}$]. **i** Scanning transmission electron microscopy (STEM) high angle annular dark field (HAADF) image with energy-dispersive X-ray spectroscopy (EDX) signals for Ca (blue) and P (yellow) of particles present after 2 h of incubation in RPMI1640/10%FBS and 2.5 mM [$Ca^{2+}$] ($n = 2$). **j** EDX-spectrum of particles shown in **i**.

nanometer-sized particles in FBS-containing medium are not visible in suspension or by light microscopy and formed only a gel-like pellet upon centrifugation. No pellet was detected in the absence of additional [$Ca^{2+}$] or when [$Ca^{2+}$] was added to low [$P_i$] media (below 3 mM [$P_i$]$_{ex}$, Supplementary Fig. 2a). Probing the isolated particles (RPMI1640/10% FBS/2.5 mM [$Ca^{2+}$]) by Western blotting confirmed, that they indeed contain fetuin-A (Fig. 1f).

Visualization of the particles with transmission electron microscopy (TEM) and cryo-TEM confirmed their size distribution and revealed a non-crystalline, amorphous, colloidal structure (Fig. 1g, h). The comparative analysis of particle size by TEM and Cryo-TEM was performed in order to ensure that drying particles on TEM-grids does not alter their shape or size. Subsequently, TEM combined with energy-dispersive X-ray spectroscopy (EDX) was used to further characterize the

elemental composition of the spontaneously formed particles (Fig. 1i, j). EDX analysis revealed roughly equal amounts of calcium and phosphorus within the first 4 h (Fig. 1j) and a slight excess of calcium over phosphorus after 8 h, which is accompanied by minor increases in particle size (not shown). Such particles have been described as colloidal spheres containing fetuin-A, calcium, and phosphorus and are referred to as CPPs[33,34].

To confirm the presence of CPPs in monocytes stimulated with [$Ca^{2+}$]$_{ex}$, TEM was used again. 30 min after addition of Ca$^{2+}$, electron-dense particles were visible by TEM and scanning transmission electron microscopy (STEM) in the extracellular space close to the cell surface, but also in the cytoplasm (Fig. 2a, b). The particles correspond in size and elemental composition to the CPPs detected in cell-free, FBS-supplemented RPMI with 2.5 mM [$Ca^{2+}$]. EDX analysis showed that those particles contain high

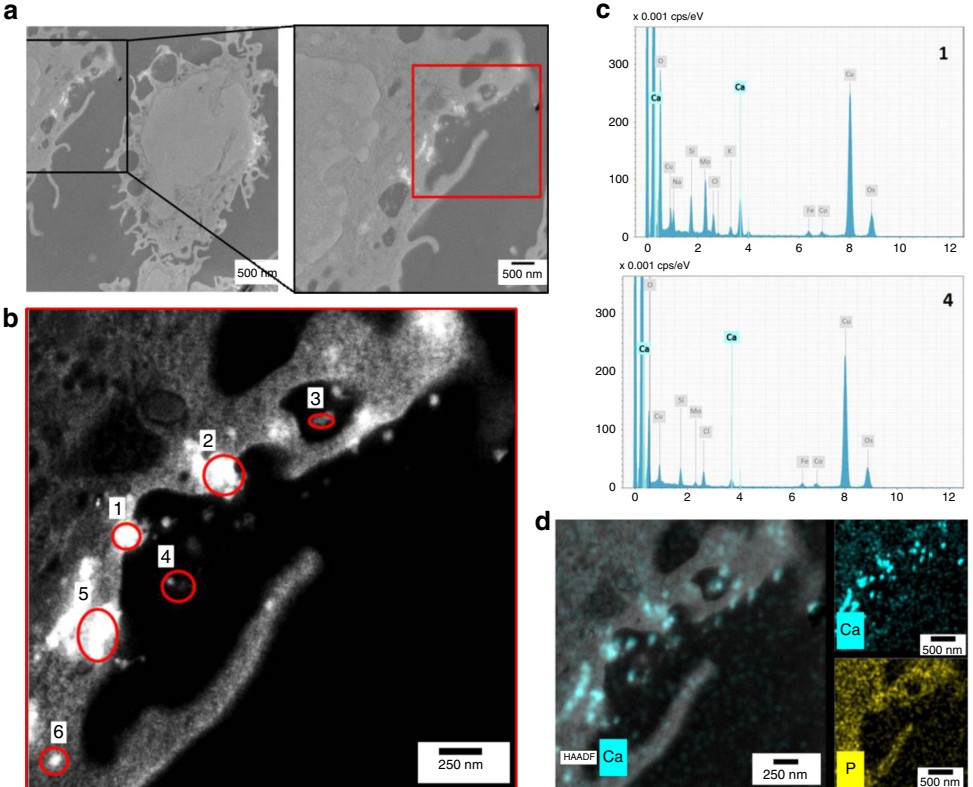

**Fig. 2 Uptake and elemental composition of intracellular calciprotein particles. a–d** Human freshly isolated blood monocytes were stimulated with 2.5 mM added $[Ca^{2+}]$ in 5.6 mM $[P_i]$ for 30 min, fixed, and embedded for TEM–EDX analysis ($n = 2$). **a** STEM HAADF image and higher magnification of the region framed in black; bright areas are electron dense regions. **b** STEM HAADF imaging of the red framed region in **a**. **c** EDX signals for Ca and P. EDX spectra of regions circled in red indicate that bright areas are containing Ca, as indicated by the calcium peaks in the two representative EDX spectra for regions 1 and 4 in **b** (for spectra from the other marked areas, see Supplementary Fig. 2b). The phosphor signal at 2 keV overlaps with the osmium signal which is found to be in the same region of the spectra. For this reason, both signals are not considered here and the region was excluded during spectra acquisition and mapping of cell samples. **d** STEM HAADF image of **b** with EDX signals for Ca and P.

amounts of $Ca^{2+}$, indicating that they are indeed identical to the CPPs described above, both, in size and in composition (Fig. 2c, d; Supplementary Fig. 2b).

**CPPs are taken up by CaSR-mediated macropinocytosis.** Increased $[Ca^{2+}]_{ex}$ stimulates macropinocytosis, which led us to hypothesize, that spontaneously formed CPPs are engulfed by monocytes due to $[Ca^{2+}]_{ex}$-induced CaSR signaling. To test this hypothesis, we investigated $[Ca^{2+}]_{ex}$-induced macropinocytosis of monocytes using flow cytometry and imaging flow cytometry analysis. In initial experiments, we used an established protocol for the quantification of macropinocytosis, in which the fluorescent dye calcein is added to the culture medium and is then taken up by monocytes in macropinosomes containing extracellular fluid. This experiment shows that macropinocytosis in monocytes is regulated by $[Ca^{2+}]_{ex}$ in a concentration-dependent manner, but is independent of $[P_i]_{ex}$ (Fig. 3a, b).

$[Ca^{2+}]_{ex}$-induced uptake of calcein by macropinocytosis is an active process, which was abrogated at 4 °C (Fig. 3c) and could be inhibited by pre-incubation with cytochalasin D or latrunculin A (Fig. 3d, e), two inhibitors of actin polymerization. $Ca^{2+}$-induced macropinocytosis of monocytes was indeed triggered by CaSR signaling, since Calhex231 and NPS2143, which are specific negative allosteric modulators of the CaSR, inhibited it (Fig. 3f).

To confirm $[Ca^{2+}]_{ex}$-induced intracellular uptake of CPPs, live imaging of monocytes was performed by confocal Raman microspectroscopy (CRM, Fig. 3g, h). The Raman spectrum of

CPPs in cell-free medium (Fig. 3h, left) revealed a fingerprint that strongly resembled hydroxyapatite crystalline particles[35,36]. The peaks at 959, 1040, 430, and 575 $cm^{-1}$ were assigned to the symmetric/asymmetric stretching and bending modes of the $PO_4^{3-}$ group of calcium phosphate particles, respectively (Fig. 3h). In contrast to hydroxyapatite crystals, there were also additional peaks at 1450 and 1670 $cm^{-1}$ originated from the presence of proteins in the particles. The detected β-sheet amide I peak at 1670 $cm^{-1}$ was expected for fetuin proteins[37].

For the CRM live-imaging analysis, monocytes were incubated in RPMI culture medium containing 2.5 mM $[Ca^{2+}]$ and 5.6 mM $[P_i]$. After 60 min, CPPs were found in close vicinity of the cell membrane as well as internalized in the monocytes (Fig. 3g). The Raman spectrum extracted from CPPs within the cell (Fig. 3h, right) showed the presence of symmetrical stretching and bending modes of $PO_4^{3-}$ vibrations in the cell. Additional cell-specific signals from pyrrole breathing in cytochromes (1130 and 1585 $cm^{-1}$), $CH_2$ deformations of lipids and proteins (1450 $cm^{-1}$) and amide I band of proteins (1660 $cm^{-1}$) were also detected.

Next, the uptake of CPPs by macropinocytosis was visualized using imaging flow cytometry by incubating monocytes with preformed CPPs stained with fluorescent calcein, which binds to $Ca^{2+}$. Fluorescence images showed calcein fluorescence in intracellular areas, which likely represent dense accumulation of CPPs, since they are considerably larger than individual particles. 2.5 mM $[Ca^{2+}]_{ex}$ triggered significant uptake of CPPs into the monocytes, while only very low uptake was detectable without additional $[Ca^{2+}]_{ex}$ (Fig. 4a, for gating strategy applied see

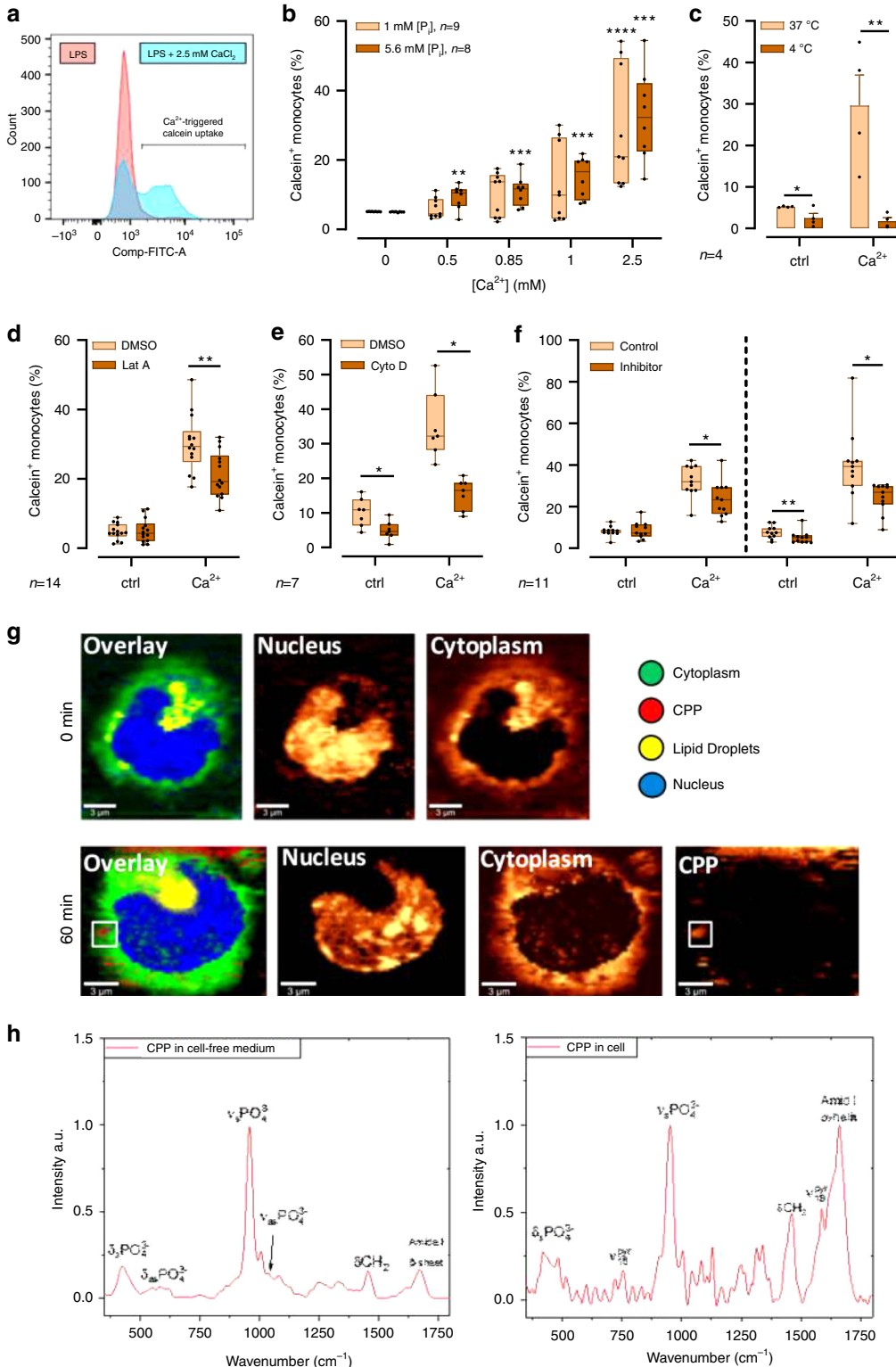

Supplementary Fig. 2c, d). When $[Ca^{2+}]$ was replaced by barium, another known CaSR ligand, similar rates of CPP uptake were detectable (Fig. 4b).

Finally, $[Ca^{2+}]_{ex}$-induced uptake of calcein-stained CPPs was quantified in CaSR-deficient THP-1 cells by imaging flow cytometry and was found to be significantly lower compared to control cells (Fig. 4c).

To further investigate CaSR-mediated effects of $[Ca^{2+}]_{ex}$ and $[P_i]_{ex}$ on monocytic cells, we performed label-free dynamic mass redistribution (DMR; Corning Epic System) measurements in monocytes. CaSR has been reported to couple to several G proteins including $G_{i/o}$, $G_{q/11}$, and $G_{12/13}$[38]. In comparison to performing several second messenger assays, the DMR measurements provide the advantage that receptor activation is measured independently of the activated signaling cascades. The sum of all cellular responses is recorded in a time-resolved manner.

The $[Ca^{2+}]_{ex}$-induced and $[P_i]_{ex}$-induced response curves showed a fast cellular response to addition of 1 mM $[Ca^{2+}]$ in

**Fig. 3 [Ca²⁺]ₑₓ induces macropinocytosis. a–f** Measurement of calcein uptake by flow cytometric analysis. Freshly isolated human blood monocytes were incubated with LPS and 2.5 mM added [Ca²⁺] for 45 min in calcein-stained (15 μM) RPMI1640/10% FBS containing 5.6 mM [Pᵢ], if not indicated otherwise. Inhibitors were pre-incubated for 30 min in calcein-free medium. 20,000 cells were detected in each experiment. Box-and-Whisker plots show median, 25–75th percentile, and min/max whiskers, of experiments with the indicated numbers of donors. Statistical analysis was performed using two-tailed Mann–Whitney U test **b**, t-test **c**, Wilcoxon signed-rank test. **d–f** p-values are indicated as *p < 0.05, **p < 0.01, ***p < 0.001, ****p < 0.0001. Representative histogram and analysis strategy for flow cytometric measurement **a** and quantification of monocytic [Ca²⁺]-dependent calcein uptake in 1 or 5.6 mM [Pᵢ] medium **b**, quantification of calcein uptake at 37 vs. 4 °C **c**, after pre-incubation with 2 μM Latrunculin A (Lat A) **d**, 1 μM Cytochalasin D (Cyto D) **e**, 10 μM Calhex231 or 10 μM NPS2143 **f** or the corresponding DMSO concentration prior to [Ca²⁺]-stimulation. **g, h** Monitoring of CPP formation and their internalization by human monocytes using live-imaging-confocal Raman microspectroscopy (CRM). CRM live cell imaging of one monocyte prior to [Ca²⁺] treatment (0 min) and 60 min after addition of 2.5 mM [Ca²⁺] to RPMI1640 medium (bottom) **g**. The color-coded images (left) represent the overlay of imaging of cytoplasm, nucleus, lipid droplets, and CPPs. The single images visualize nucleus, cytoplasm, and CPP separately, as indicated. Raman spectra of CPPs in cell-free RPMI medium containing 10% FBS and 2.5 mM [Ca²⁺] (left) and after their internalization into the monocyte (right) **h**. Characteristic Raman peaks are assigned in the spectra (pyr pyrrole rings of porphyrin, d deformation, s symmetric, as asymmetric). Shown is one cell out of 10 analyzed in experiments with monocytes from four donors.

control cells, which was significantly reduced in CaSR-deficient THP-1 cells (def-CaSR, Fig. 4d). Nevertheless, CaSR-deficient cells still showed some response to [Ca²⁺] addition, most likely due to CaSR independent effects.

The [Ca²⁺]ₑₓ-induced and [Pᵢ]ₑₓ-induced DMR measurements were also performed in primary monocytes. The response was detectable after 2–5 min, started to taper off after 10–15 min and showed a secondary slower increase after 30 min (Fig. 4e). Monocytic DMR responses to increased [Ca²⁺]ₑₓ were profoundly influenced by [Pᵢ]ₑₓ, with a concentration-dependent augmentation of the signal (Fig. 4e).

To dissect CaSR-mediated signaling from non-specific effects on monocytes, the Gαq inhibitor YM254890 was used, which had a significant influence on the DMR response at 5.6 mM [Pᵢ]ₑₓ, but not at 1 mM [Pᵢ]ₑₓ (Fig. 4f). Importantly, the early peak of the DMR signal during the first 20 min after stimulation, which was highest in the presence of 2.5 mM [Ca²⁺]ₑₓ and 5.6 mM [Pᵢ]ₑₓ, was inhibited by YM254890. This confirmed that a [Ca²⁺]ₑₓ-sensitive Gαq protein-coupled receptor is transmitting the signal into the monocytes (Fig. 4f).

Uptake of material into monocytes by endocytosis is associated with subsequent merging of the endosomes with lysosomes and proteolytic breakdown of proteins by cathepsins. To detect lysosomal activity following CaSR-mediated uptake of CPPs into the monocytes, cathepsin B activity was determined. Addition of [Ca²⁺] to 5.6 mM [Pᵢ]-containing RPMI1640/10% FBS, which triggers spontaneous formation of CPPs, also led to a [Ca²⁺]-dependent increase in cathepsin B activity (Fig. 4g). In contrast, no [Ca²⁺]ₑₓ-induced increase was detectable at low [Pᵢ]ₑₓ.

**IL-1β triggered by CPPs and [Ca²⁺]ₑₓ is [Pᵢ]ₑₓ independent.** To investigate the role of CPPs in [Ca²⁺]ₑₓ-induced NLRP3 inflammasome activation, monocytes were stimulated with [Ca²⁺]ₑₓ in the presence of preformed CPPs in low [Pᵢ] medium to rule out de novo formation of CPPs. In the presence of 2.5 mM [Ca²⁺]ₑₓ, CPPs were found to trigger a concentration-dependent IL-1β release despite the low [Pᵢ]. CPPs had only a minor effect on IL-1β release if no further Ca²⁺ was added (Fig. 5a). Comparable to the experiments shown in Fig. 1, monocytes did not produce IL-1β after addition of Ca²⁺ at low [Pᵢ] in the absence of CPPs (Fig. 5a).

Kinetic analysis of monocytic IL-1β release following [Ca²⁺]ₑₓ-induced CPP uptake showed, that addition of CPP in low [Pᵢ] medium in the presence of [Ca²⁺]ₑₓ started already after 3 h and was faster compared to addition of [Ca²⁺] in the presence of 5.6 mM [Pᵢ] (Fig. 5b).

[Ca²⁺]ₑₓ-induced IL-1β release was dependent on actin polymerization in monocytes, because cytochalasin D and latrunculin B inhibited it (Fig. 5c, Supplementary Fig. 2e). The

hypothesis that lysosomal digestion of engulfed CPPs contributes to the [Ca²⁺]ₑₓ-induced inflammasome activation and IL-1β release in monocytes was further supported by the observation, that pharmacological cathepsin B inhibition with CA-074-Me decreased the [Ca²⁺]-induced IL-1β response (Fig. 5d).

When 2.5 mM [Ca²⁺] was added to RPMI1640/10% FBS containing high [Pᵢ], and the medium subsequently passed through a 100-nm pore-size filter, the mean particle size determined by DLS was reduced by 20 nm (not shown) and the particle amount was decreased as indicated by the reduced sedimentation shown in Fig. 5e. This filtration almost abrogated the [Ca²⁺]ₑₓ-induced IL-1β production (Fig. 5e), indicating that size and concentration of CPPs were relevant for inflammasome activation.

When the ratio between fetuin-A and [Ca²⁺] during particle generation was increased by raising fetuin-A concentrations in the culture medium, a concentration-dependent decrease of IL-1β release was observed (Fig. 5f). The pivotal role of CPP formation was confirmed by experiments with phosphonoformic acid (PFA), an inhibitor of calcium phosphate crystallization. Addition of PFA inhibited concentration dependently the [Ca²⁺]ₑₓ-induced IL-1β release (Fig. 5g).

To formally exclude an unspecific, particle-induced stimulatory effect as the underlying mechanism of IL-1β release, albumin-coated BaSO₄ nanoparticles with characteristics and size similar to CPPs were produced. These nanoparticles had no effect on LPS-primed monocytes, neither in the absence nor in the presence of increased [Ca²⁺]ₑₓ (Fig. 5h). When BaSO₄ nanoparticles were added to LPS-primed monocytes in parallel to stimulation with MSU crystals, they did also not increase the induced IL-1β response (Supplementary Fig. 2f). In monocytes primed with LPS for 6 h and stimulated with ATP for 30 min, there was a dose-dependent trend for BaSO₄ nanoparticles to inhibit IL-1β release, which did not reach statistical significance (Supplementary Fig. 2g).

Lastly, we investigated CPP uptake and [Ca²⁺]ₑₓ-induced IL-1β release of monocytes after incubation in high [Pᵢ] medium in a cohort of healthy human donors and found considerable variation. Strikingly, a close correlation between macropinocytosis and the induced IL-1β was discernible, further corroborating the link between CPP uptake and inflammasome activation (Fig. 5i).

**[Ca²⁺]ₑₓ-induced IL-1β response in RA is increased.** The in vivo relevance of [Ca²⁺]ₑₓ-induced, CaSR-mediated IL-1β release has initially been shown in patients with CAPS[1] and in an inflammatory mouse model[2]. In order to further substantiate the link between in vivo inflammation and the response to [Ca²⁺]ₑₓ and CPPs, peripheral blood monocytes from patients with RA (n = 39) treated with stable disease-modifying anti-rheumatic drugs

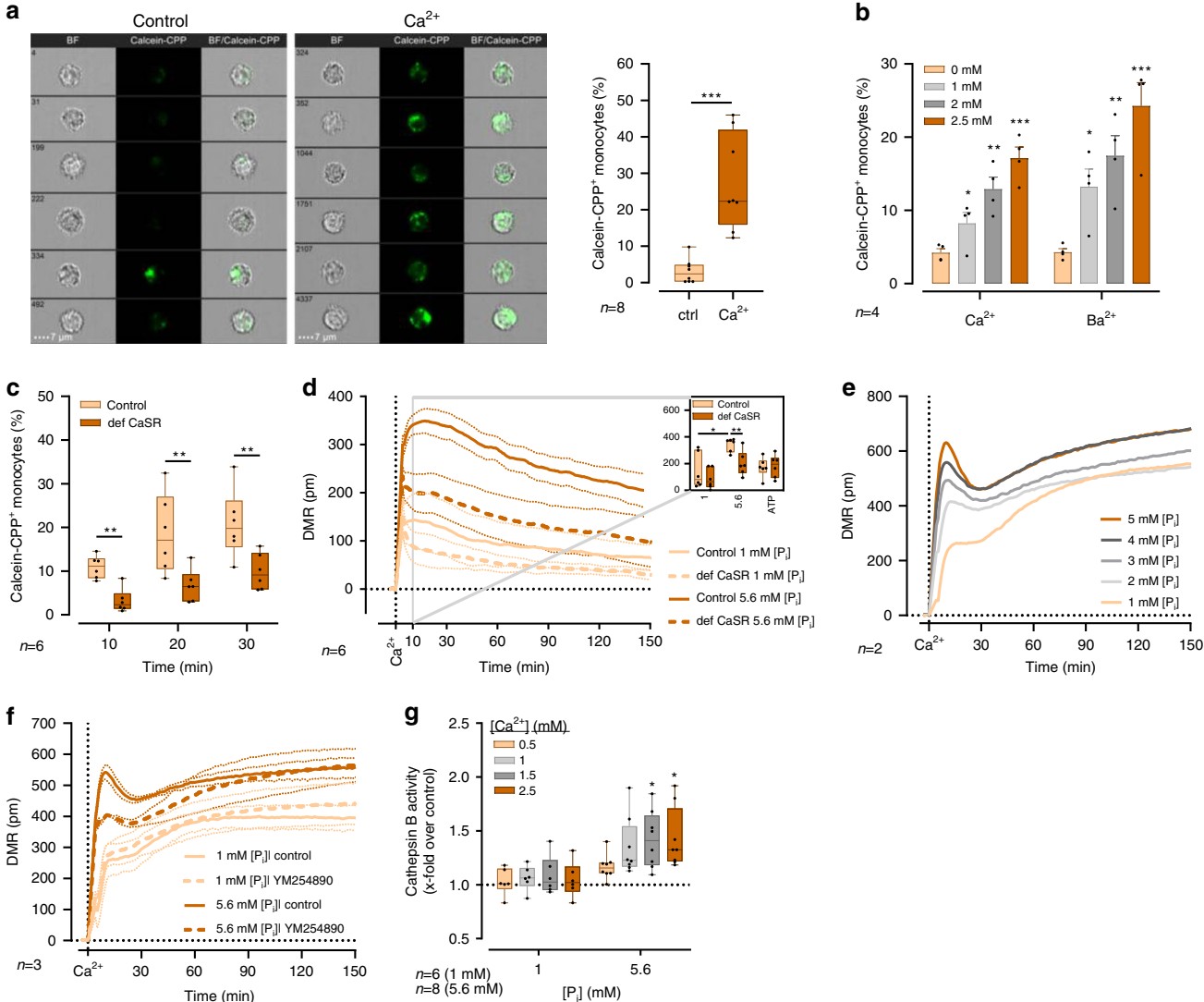

**Fig. 4 Macropinocytosis of CPPs depends on CaSR signaling. a** [Ca$^{2+}$]-dependent uptake of calcein-stained CPPs (calcein-CPP) was visualized and analyzed with ImageStreamX Mark II. 5,000–10,000 freshly isolated human blood monocytes from eight donors were imaged and analyzed for calcein fluorescence after 45 min incubation with calcein-CPPs in 1 mM [P$_i$] RPMI1640/10%FBS. Representative images are shown for monocytes incubated with calcein-CPPs in the absence (left) and presence (right) of 2.5 mM added [Ca$^{2+}$]. **b** Uptake of calcein-stained CPPs (calcein-CPP) after incubation with the indicated added concentrations of [Ca$^{2+}$], or [Ba$^{2+}$] in RPMI1640/10%FBS containing 1 mM [P$_i$] for 45 min (four donors). **c** Calcein-CPP uptake was analyzed in control and CaSR-deficient THP-1 cells (five experiments). Cells were incubated with calcein-stained CPPs and stimulated with 2.5 mM added [Ca$^{2+}$] for 10, 20, and 30 min, as specified. **d** Dynamic mass redistribution (DMR) measurement in freshly isolated blood monocytes is shown as mean ± s.e.m. (dotted curves) of six experiments from six donors. Addition of agents of interest is indicated as vertical dotted line. Cells were either stimulated in 1 or 5.6 mM [P$_i$]-containing RPMI1640/10%FBS medium as indicated. Inset shows median of time point 10 min. **e, f** DMR analysis of human monocytes stimulated with 2.5 mM added [Ca$^{2+}$] in RPMI1640/10%FBS medium containing [P$_i$] as indicated (**e**, two donors) and after pre-incubation with 5 μM of the Gα$_q$ inhibitor YM254890 or the corresponding DMSO dilution (control) for 60 min (**f**, three donors). Data are shown as mean. **g** Cathepsin B activity determined by magic red fluorescence in monocytes from the indicated numbers of donors stimulated with increasing [Ca$^{2+}$] in RPMI1640/10% FBS containing 1 or 5.6 mM [P$_i$] is shown as x-fold over control (LPS); Box-and-Whisker plots show median, 25th–75th percentile, and min/max whiskers, bar charts show mean ± s.e.m., unless indicated otherwise. Statistical analysis was performed using two-tailed Mann–Whitney $U$ test (**a, c, d**) or two-tailed $t$-test **b**. $p$-values are indicated as *$p < 0.05$, **$p < 0.01$, ***$p < 0.001$.

(DMARD) therapy (conventional synthetic (cs) DMARD, biological (b) DMARD, or targeted synthetic (ts) DMARD, for details see section "Methods") and in partial remission were investigated. Stimulation with increasing [Ca$^{2+}$]$_{ex}$ led to higher IL-1β release in RA monocytes than in healthy donors (Fig. 6a). To determine the influence of treatment and disease activity on [Ca$^{2+}$]$_{ex}$-induced IL-1β release, a cohort of patients without DMARD ($n = 12$) was investigated, which in part ($n = 7$) consisted of patients with recent-onset RA previously not treated with DMARD or glucocorticoids. Disease activity in this cohort was significantly higher than in patients treated with DMARD (median disease activity score DAS28 3.68 vs. 2.91, $p = 0.01$). Figure 6a shows that RA patients without DMARD produced even higher IL-1β concentration upon stimulation with [Ca$^{2+}$]$_{ex}$. In addition to IL-1β, RA monocytes also released higher concentrations of IL-1α and IL-18 upon stimulation with [Ca$^{2+}$]$_{ex}$ compared to healthy donors (Fig. 6b, c).

To determine the specificity of the observed increase of [Ca$^{2+}$]$_{ex}$-induced IL-1β release for RA, a cohort of psoriatic arthritis patients (PsA, $n = 8$) and a cohort of patients with

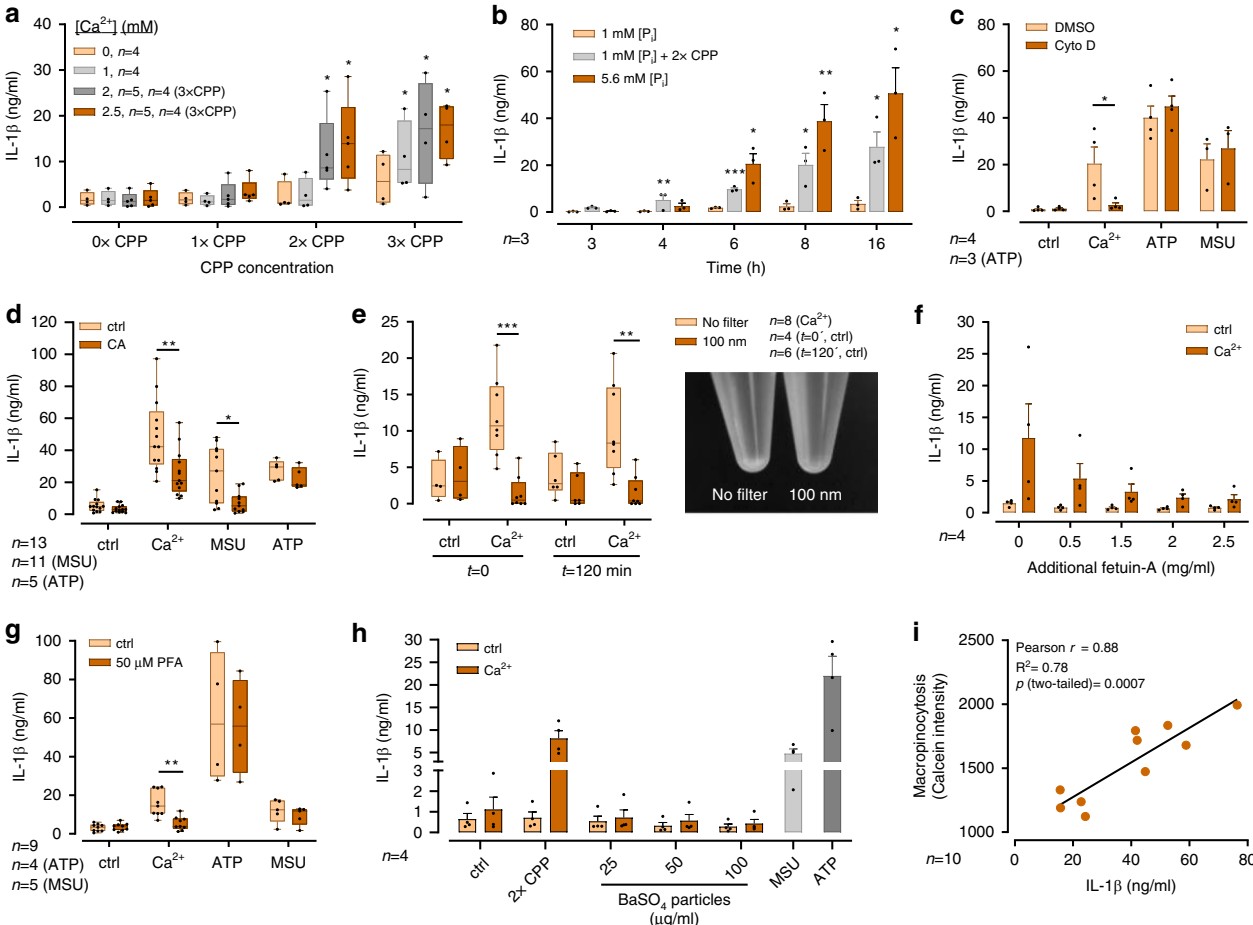

**Fig. 5 [Ca²⁺]ₑₓmediated calciprotein particle uptake mediates inflammasome activation. a–h** Detection of IL-1β via ELISA in cell culture supernatants of human freshly isolated peripheral blood monocytes from the indicated number of donors after 16 h of incubation with LPS (100 ng/ml) and the indicated substances. **a** Treatment with different concentrations of CPPs and the indicated [Ca²⁺] in RPMI1640/10%FBS containing 1 mM [Pᵢ]. **b** Time course of IL-1β release after addition of 2.5 mM [Ca²⁺] to monocytes in 1 mM [Pᵢ] plus CPPs compared to monocytes in RPMI1640/10%FBS containing 5.6 mM [Pᵢ]. **c** Pre-incubation with 1 μM Cytochalasin D (Cyto D) or DMSO prior to stimulation with 2.5 mM [Ca²⁺] or 3 mM ATP or 100 μg/ml MSU in RPMI1640/10%FBS containing 5.6 mM [Pᵢ]. **d** Pre-incubation with 3 μM CA-074-Me or DMSO for 30 min prior to stimulation with 2.5 mM added [Ca²⁺] or 3 mM ATP or 100 μg/ml MSU in RPMI1640/10%FBS containing 5.6 mM [Pᵢ]. **e** Incubation in filtrated (100 nm) or non-filtrated (no filter) RPMI1640/10%FBS with 2.5 mM added [Ca²⁺]. Medium was either pre-incubated with [Ca²⁺] for 120 min prior to filtration (t = 120 min) or directly filtrated (t = 0) through a 100 nm pore-size filter. Visualization of particle reduction after filtration is shown in the graph inset, particulate matter was pelleted by 2 h centrifugation at 16,000 × g. **f** Incubation in RPMI1640/10%FBS containing 5.6 mM [Pᵢ] with additional fetuin-A concentrations and stimulation with 2.5 mM added [Ca²⁺]. **g** Addition of phosphonoformic acid (PFA) into RPMI1640/10%FBS containing 5.6 mM [Pᵢ] prior to stimulation with 2.5 mM added [Ca²⁺] or 3 mM ATP or 100 μg/ml MSU. **h** Treatment of monocytes with the indicated concentrations of albumin-coated BaSO₄ nanoparticles in the absence (ctrl) or presence of 2.5 mM added [Ca²⁺] or 3 mM ATP or 100 μg/ml MSU. **a–h** Box-and-Whisker plots show median, 25–75th percentile, and min/max whiskers, bar charts show mean ± s.e.m. Statistical analysis was performed using two-tailed Mann–Whitney U test (**a, d, e, g**) or two-tailed t-test **b, c**. p-values are indicated as *p < 0.05, **p < 0.01, ***p < 0.001. **i** Correlation between IL-1β release (16 h) and macropinocytosis (calcein uptake after 45 min) of monocytes from 10 healthy donors after stimulation with LPS and 2.5 mM added [Ca²⁺] in RPMI1640/10%FBS containing 5.6 mM [Pᵢ]. Two-tailed Pearson correlation coefficient and level of significance as indicated.

systemic lupus erythematodes (SLE, n = 5) were investigated. The results shown in Fig. 6a–c indicate, that in those two diseases, monocytes released significantly lower concentrations of IL-1β, IL-1α, and IL-18 upon [Ca²⁺]ₑₓ stimulation compared to RA patients without DMARD treatment (Fig. 6a). PsA and SLE monocyte responses did not differ significantly from healthy donors or from RA patients on stable DMARD treatment, but further studies of larger cohorts are required for final assessment. MSU crystal-induced monocytic IL-1β and IL-1α release was comparable in all diseases investigated and did not differ from healthy donors (Supplementary Fig. 3a, b).

Importantly, in the presence of 1.1 mM [Ca²⁺]ₑₓ (labeled as 0.85 mM added Ca²⁺), which is within the concentration range

present in rheumatoid synovial fluid, RA monocytes also produced significantly more IL-1β in response to preformed CPPs (Fig. 6d). No difference was discernible at higher [Ca²⁺]ₑₓ (1.7 mM, Fig. 6d). When monocyte-derived macrophages were differentiated in vitro from RA monocytes, they also responded with higher IL-1β concentrations after stimulation with [Ca²⁺]ₑₓ compared to monocyte-derived macrophages from healthy donors (Fig. 6e).

To determine the influence of RA on [Ca²⁺]ₑₓ-induced CPP uptake, monocytes from RA patients were analyzed for CPP macropinocytosis by imaging flow cytometry. Figure 6f shows, that compared to healthy donors, higher percentages of RA monocytes internalize calcein-stained, preformed CPPs, indicating

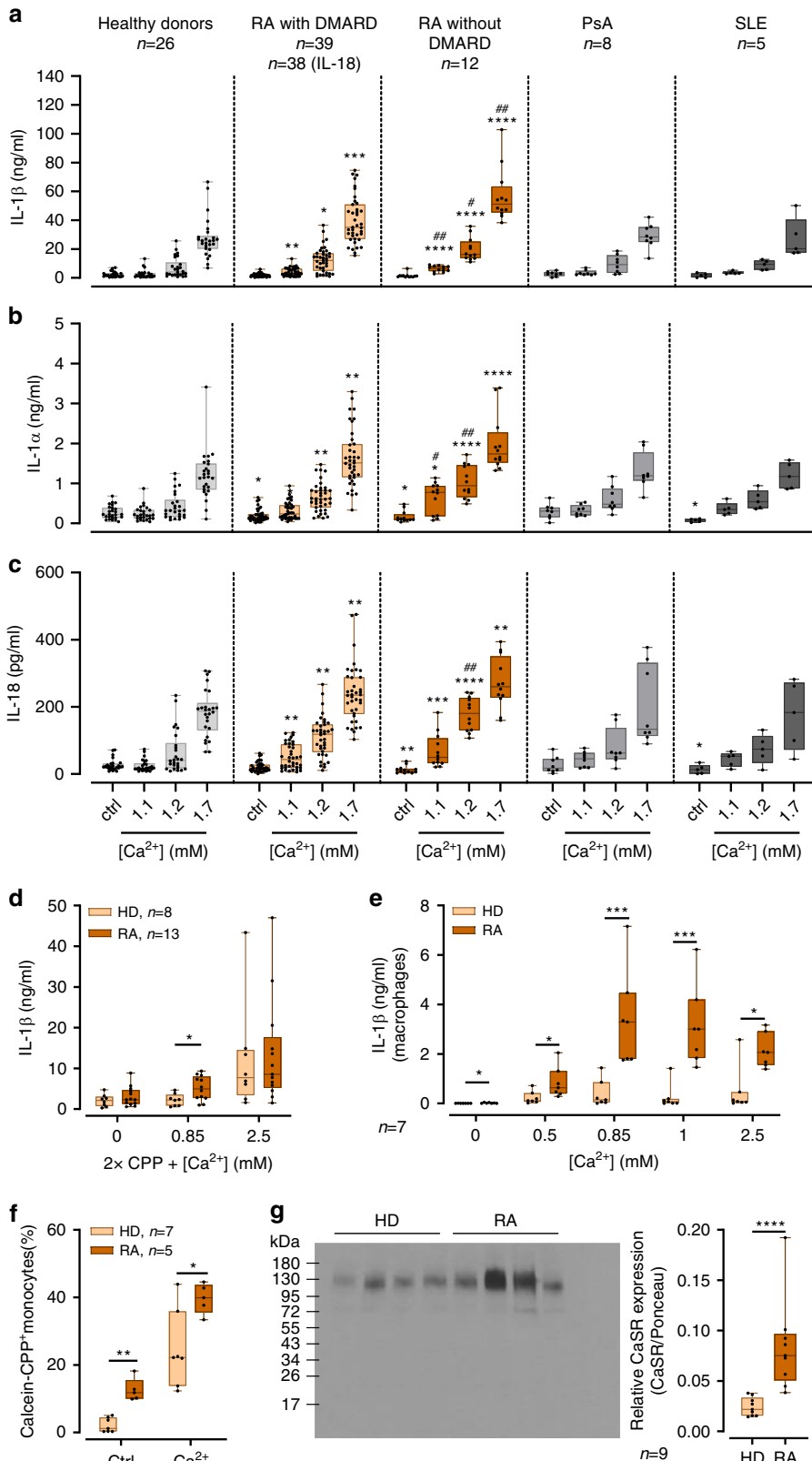

their increased propensity to respond to $[Ca^{2+}]_{ex}$, but also to LPS alone.

We hypothesized, that the observed increase in $[Ca^{2+}]_{ex}$-induced IL-1β release and CPP uptake in RA could result from increased CaSR signaling. Therefore, peripheral blood monocytes from RA patients were analyzed for CaSR expression by Western

blot, and showed indeed significantly higher expression levels than healthy donors (Fig. 6g, ponceau S staining of membrane in Supplementary Fig. 3c).

In the past, RA and other autoimmune diseases have been associated with increased activity of lysosomal hydrolases[39,40] and decreased lysosomal membrane stability (reviewed in ref. [41]). To

**Fig. 6 Increased [Ca$^{2+}$]$_{ex}$-induced, CPP-dependent inflammasome activation in RA. a–c** Detection of IL-1β **a**, IL-1α **b**, and IL-18 **c** in cell culture supernatants of freshly isolated peripheral blood monocytes from patients with RA (RA), psoriatic arthritis (PsA), systemic lupus erythematodes (SLE), or healthy donors (HD) after 16 h of incubation with LPS and [Ca$^{2+}$] in RPMI1640/10%FBS/5.6 mM [P$_i$] media. *Indicates level of significance for comparison against the same concentration in healthy donors; #for comparison between RA with and without DMARD. Comparison to RA without DMARD in panel **a**: $p = 0.0073$ (1.2 mM) and $p < 0.0001$ (1.7 mM) for PsA, $p = 0.0013$ (1.2 mM) and $p = 0.0094$ (1.7 mM) for SLE. **d** Stimulation of freshly isolated peripheral blood monocytes from RA patients and healthy donors (HD) with 2x CPPs and the indicated added [Ca$^{2+}$] in 1 mM [P$_i$] RPMI1640/10%FBS. **e** Stimulation of human monocyte-derived macrophages from RA patients (RA) or healthy donors (HD) with the indicated added [Ca$^{2+}$] and detection of IL-1β in cell culture supernatants after 16 h of incubation in 5.6 mM [P$_i$] RPMI1640/10%FBS; **f** Calcein-CPP uptake of human monocytes from either RA patients (RA) or healthy donors (HD) after stimulation with 2.5 mM added [Ca$^{2+}$] for 45 min, detection and quantification was done by imaging flow cytometry (ImageStream$^X$MarkII); **g** Representative Western blot and dot plot showing expression levels of CaSR in freshly isolated monocytes from either healthy donors (HD) or RA patients (RA), Ponceau staining of PVDF membranes was used for quantification of CaSR protein (see Supplementary Fig. 3c). Box-and-whisker plots show median, 25–75th percentile, and min/max whiskers. Patient numbers are indicated in the figure. Statistical analysis was performed using two-tailed Mann–Whitney $U$ test. $p$-values are indicated as *$p < 0.05$, **$p < 0.01$, ***$p < 0.001$, ****$p < 0.0001$.

investigate whether lysosomal leakage contributes to [Ca$^{2+}$]$_{ex}$-induced NLRP3 inflammasome activation, lysosomes were stained with acridine orange and stimulated either with 2.5 mM [Ca$^{2+}$] in the presence of 5.6 mM [P$_i$]$_{ex}$, or with MSU crystals or the lysosomal disrupting agent L-leucyl-L-leucine methyl ester (LLOMe) as positive controls[42]. Imaging flow cytometry showed, that lysosomal membrane integrity did not change after stimulation with [Ca$^{2+}$]$_{ex}$ compared to the negative control (Fig. 7a). In contrast, MSU crystals and LLOMe induced lysosomal leakage, as indicated by a significantly reduced bright detail fluorescence intensity after 4 h of incubation. Comparison of lysosomal leakage between RA monocytes and healthy donors showed no differences (Fig. 7a).

Inflammasome activation and subsequent IL-1β release are commonly thought to be associated with pyroptotic or inflammatory cell death, which prompted us to investigate cell survival. Both, propidium iodide (PI)/Hoechst staining analyzed by celigo image cytometer (Fig. 7b, c) and standard LDH release assays (Fig. 7d), indicated elevated cell death under conditions where IL-1β release occurred. Monocytes from RA patients showed increased cell survival upon stimulation with LPS alone, but their cell death rate induced by [Ca$^{2+}$]$_{ex}$ or ATP did not differ from healthy donors (Fig. 7d). In the healthy donor cohort described above, cell death rates at elevated [Ca$^{2+}$]/[P$_i$] conditions strongly correlated with the corresponding IL-1β release (Fig. 7e). Cell death was partially dependent on the presence of NLRP3 (Fig. 7f) and CaSR (Fig. 7g).

It has been suggested, that cell death—independent of IL-1β maturation and release—contributes to pathological damage in MSU-induced gouty arthritis[43]. In order to investigate how inhibiting [Ca$^{2+}$]$_{ex}$-induced inflammasome activation either upstream or downstream of NLRP3 influences cell death, inhibitors of CaSR, cathepsin B, and caspase-1 were used. The results indicate, that inhibition of CaSR-mediated CPP uptake or of lysosomal CPP breakdown lead to a significant inhibition of cell death (Fig. 7h).

**[Ca$^{2+}$]$_{ex}$-induced IL-1β contributes to local inflammation.** We next investigated the influence of arthritis on [Ca$^{2+}$] and [Ca$^{2+}$]$_{ex}$-induced IL-1β secretion. Synovial fluid samples from patients with RA contained higher [Ca$^{2+}$] than joint effusions from patients with osteoarthritis or non-erosive joint diseases (Fig. 8a). Staining of RA synovial membrane cryostat sections with Calcium Red showed accumulation of Ca$^{2+}$ in the sub-synovial lining, which could be removed by pre-treatment of the sections with ethylenediaminetetraacetic acid (EDTA) (Fig. 8b). In addition, CaSR expression was found to be upregulated in the synovial lining layer in RA patients compared to osteoarthritis samples (Fig. 8c). To investigate the role of CaSR in arthritis in vivo, the allosteric CaSR modulator R568 was used in the

collagen-antibody-induced arthritis (CAIA) model in DBA/1J mice. The result shown in Fig. 8d indicate, that positive modulation of CaSR signaling aggravates arthritis in this mouse model.

To further investigate the role of [Ca$^{2+}$] in erosive arthritis, the collagen-induced arthritis (CIA) model in DBA/1J mice was used. First, CD11b$^+$ mononuclear cells were isolated from peripheral blood, bone marrow, and the peritoneal cavity to confirm the link between erosive arthritis and the monocytic IL-1β response to increased [Ca$^{2+}$]$_{ex}$. In all investigated cell populations, cells from CIA mice showed a stronger [Ca$^{2+}$]$_{ex}$-induced IL-1β response compared to cells from control mice (Fig. 8e). Similarly, secretion of IL-6 was also found to increase with increasing concentrations of [Ca$^{2+}$]$_{ex}$ plus LPS, and to be higher in mononuclear cells from mice with CIA compared to control mice (Supplementary Fig. 3d).

Calcium Red staining of cryostat sections of paws showed positive staining at sites of bone erosions and on cartilage-deprived articular surfaces (Fig. 8f). Therefore, the release of Ca$^{2+}$ from bone in CIA was determined by purging bone marrow from the femur and measuring [Ca$^{2+}$] in these bone marrow flushes. Arthritic DBA/1J mice had significantly increased intramedullary [Ca$^{2+}$] compared to control mice (Fig. 8g), indicating elevated liberation of Ca$^{2+}$ from the bone matrix in CIA. The [Ca$^{2+}$]$_{ex}$-induced release of IL-1β correlated with both the amounts of [Ca$^{2+}$] determined in the bone marrow flushes (Fig. 8h) and the severity of arthritis as determined by the CIA score (Fig. 8i).

## Discussion

In the presence of fetuin-A and elevated phosphate, increased [Ca$^{2+}$] leads to CPP formation. We show here that [Ca$^{2+}$] concentrations exceeding the fetuin-A-binding capacity trigger uptake of primary CPPs into monocytes via CaSR signaling, with the consequence of NLRP3 inflammasome activation. The threshold [Ca$^{2+}$]$_{ex}$ concentration sufficient to trigger IL-1β release in the presence of a high CPP load can be as low as 1.1 mM in vitro, which is thought to be close to the physiological range in tissue and is also the concentration we found in RA synovial fluid[12]. Approximately 50% of the [Ca$^{2+}$]$_{ex}$-induced IL-1β release is mediated by CaSR, while NLRP3 inhibition or knock-down leads to almost complete abrogation of IL-1β release. The results also confirm involvement of a CaSR-independent cellular mechanism to [Ca$^{2+}$]$_{ex}$-induced IL-1β release.

Physiologically, [Ca$^{2+}$]$_{ex}$-induced activation of monocytes and NLRP3 inflammasome assembly is likely to occur primarily not in the systemic circulation, but rather in localized scenarios, where rising [Ca$^{2+}$]$_{ex}$ triggers a first line of defense response of monocytes with chemotaxis[44] and inflammatory consequences. Spontaneous formation of fetuin-A-containing CPPs under conditions of high [Ca$^{2+}$] and [P$_i$] is likely to limit inflammatory

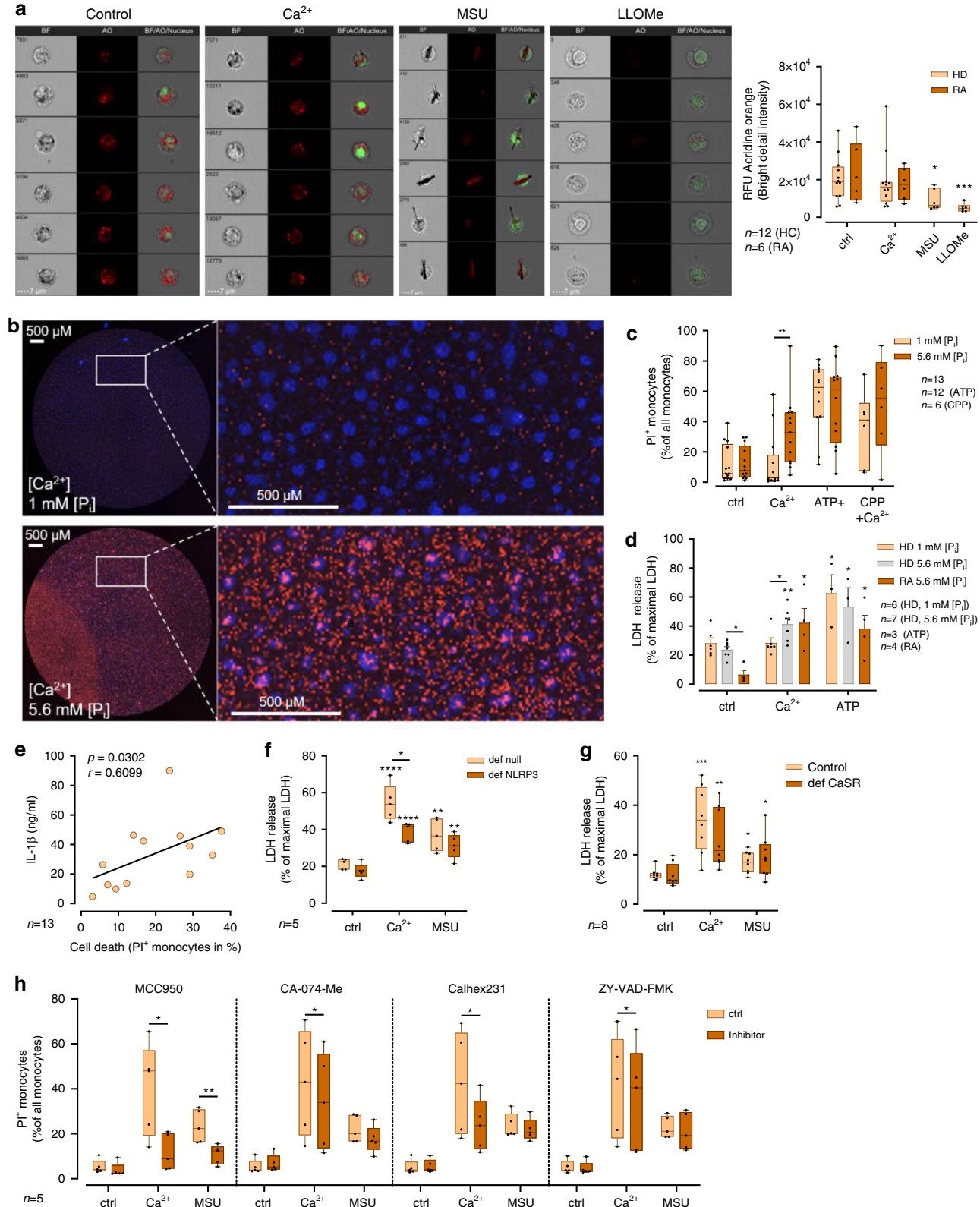

consequences as well as vascular calcifications, as long as fetuin-A is available in sufficient quantities to prevent excessive increase of $[Ca^{2+}]$ and $[P_i]$[45]. This is confirmed by our findings, that low fetuin-A concentrations are associated with exacerbated pro-inflammatory responses in vitro, while higher fetuin-A concentrations decrease the IL-1β production induced by excess CPP load.

The systemic clearance of CPPs and of excess minerals is realized by the reticuloendothelial system, specifically through liver sinusoidal endothelial cells for primary CPPs and via scavenger receptor A-expressing macrophages in liver and spleen for secondary CPPs[34]. In contrast, macropinocytotic uptake of CPPs by monocytes might be part of immune surveillance by the innate immune system as suggested recently[46].

**Fig. 7 [Ca$^{2+}$]$_{ex}$-induced CPP uptake in monocytes does not induce lysosomal leakage. a** Imaging flow cytometry of acridine orange-stained freshly isolated monocytes from healthy donors (HD) or RA patients after incubation with LPS and either 2.5 mM added [Ca$^{2+}$], 100 μg/ml MSU, or 500 μM LLOMe (10 min) for 4 h. Representative images are shown in left panels, BF brightfield, AO acridine orange, nucleus. Quantification of lysosomal acridine orange mean bright detail fluorescence intensities is shown. **b, c** Celigo image cytometer analysis of propidium iodide (PI, dead cells) and Hoechst-stained monocytes from 13 donors after stimulation with LPS ($n = 13$) and either 2.5 mM added [Ca$^{2+}$] ($n = 13$), 3 mM ATP ($n = 12$), or CPPs ($n = 6$) for 16 h at the indicated [P$_i$] levels **b**. Whole well scans of Ca$^{2+}$-treated monocytes stained with PI (red)/Hoechst (blue) from one representative donor is shown in **b**, white framed region is magnified. **c** Percentage of PI$^+$ monocytes of all monocytes (Hoechst$^+$). **d** Measurement of LDH release of monocytes from six healthy donors (HD) and four RA patients after stimulation with LPS and either 2.5 mM added [Ca$^{2+}$] or 3 mM ATP ($n = 3$) at the indicated [P$_i$]. LDH release is shown as percentage of total LDH. **e** Correlation of IL-1β and cell death determined by Celigo in 13 healthy blood donors after stimulation of monocytes with LPS and 2.5 mM added [Ca$^{2+}$] in 5.6 mM [P$_i$]. Spearman correlation coefficient and level of significance as indicated. **f, g** LDH release of NLRP3-deficient (**f**, def-NLRP3) and CaSR-deficient (**g**, def-CaSR) differentiated THP-1 cells, and their corresponding control cells stimulated with LPS and either 2.5 mM added [Ca$^{2+}$] or 100 μg/ml MSU for 8 h. LDH is shown as percentage of total LDH. **h** Celigo analysis of PI$^+$ cells after stimulation of monocytes with LPS and either 2.5 mM added [Ca$^{2+}$] or MSU for 16 h in the presence of inhibitors of NLRP3 (MCC950), cathepsin B (CA-074-Me), CaSR signaling (Calhex231), or caspase-1 (ZY-VAD-FMK). Box-and-whisker plots show median, 25–75th percentile, and min/max whiskers, bar charts show mean ± s.e.m. Two-tailed Mann–Whitney $U$ test **a–d**, **g**, two-tailed independent $t$-test **f**, or two-tailed paired $t$-test **h** were used. $p$-values are indicated as *$p < 0.05$, **$p < 0.01$, ***$p < 0.001$, ****$p < 0.0001$.

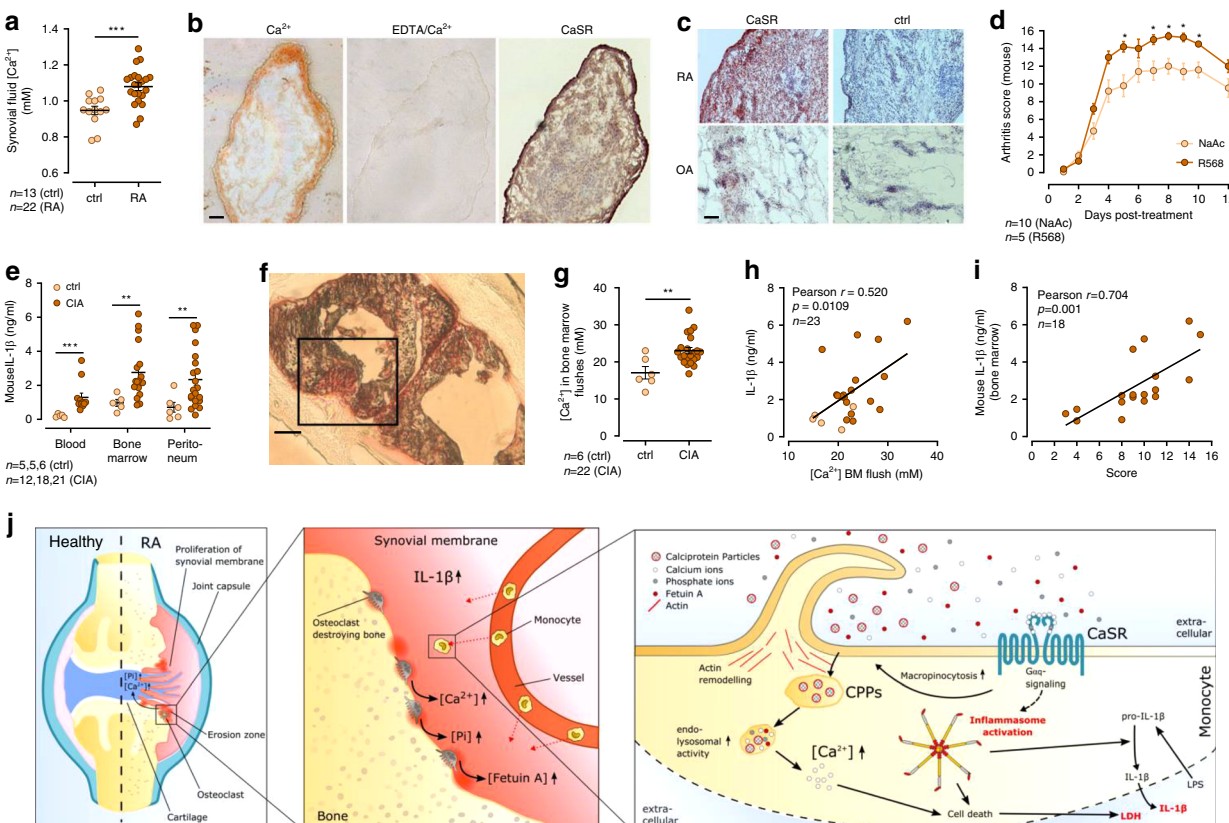

**Fig. 8 Bone erosion in arthritis leads to locally increased [Ca$^{2+}$] and elevated IL-1β secretion. a** Measurement of [Ca$^{2+}$] in synovial fluid of either RA or control patients with non-erosive arthritis or osteoarthritis (ctrl). **b** Calcium red staining of synovial membrane biopsies for Ca$^{2+}$ with Glyoxal-bis(2-hydroxyanil (GBHA) was done before or after incubation with 20% EDTA and immunohistochemistry for CaSR (right image, brown color). Scale bar = 150 μm. Shown is one representative experiment out of three. **c** CaSR protein expression was analyzed in serial sections of synovial membranes using anti-CaSR antibody (red color). Scale bar = 200 μm. Sections were counterstained with hematoxylin. Shown is one representative experiment out of six (RA) and two (osteoarthritis, OA). **d** Arthritis score in DBA/1J mice with collagen-antibody-induced arthritis (CAIA) treated i.p. with either 33 μM R568 or sodium acetate (NaAc) as control. **e** IL-1β secretion after 16 h incubation with LPS and 2.5 mM added [Ca$^{2+}$] of blood monocytes (blood), CD11b$^+$ mononuclear bone marrow cells (bone marrow) and peritoneal macrophages (peritoneum) of mice sacrificed 40 days after induction of collagen-induced arthritis (CIA) and of control animals. **f** Calcium red staining with GBHA of cryostat sections of paws from collagen-induced arthritis (CIA) mice, black frame shows region of erosion zone. Shown is one representative experiment out of six with three different mice. Scale bar = 200 μm. **g** [Ca$^{2+}$] measured in bone marrow flushes purged with 0.9% NaCl from the medullary cavity of mice sacrificed 40 days after induction of CIA or of control mice (ctrl). **h** Correlation between IL-1β release from bone marrow-derived CD11b$^+$ mononuclear cells after stimulation with LPS and 2.5 mM added [Ca$^{2+}$] for 16 h and [Ca$^{2+}$] measured in the medullary cavity from CIA mice. **i** Correlation between IL-1β release from bone marrow-derived CD11b$^+$ mononuclear cells after stimulation with LPS and 2.5 mM added [Ca$^{2+}$] for 16 h and clinical score of disease severity from CIA mice. **j** Overview of the hypothesized mechanism of Ca$^{2+}$/CPP-induced inflammasome activation in monocytes in RA. Data are shown as mean ± s.e.m. **a, d, e, g** Two-tailed Mann–Whitney $U$ test (**a, d, e, g**) or Pearson correlation coefficient (**h, i**) was used. $p$-values are indicated as *$p < 0.05$, **$p < 0.01$, ***$p < 0.001$.

MSU crystals and other crystalline salts are known to activate the NLRP3 inflammasome due to disruption of lysosomes[42]. We show here, that uptake of colloidal calciprotein nanoparticles does not induce lysosomal leakage. Increased intralysosomal $[Ca^{2+}]$ content following CPP degradation could lead to a rise of intracellular $[Ca^{2+}]$ and inflammasome assembly by gated calcium flux through members of the mucolipin subfamily of transient receptor potential channels (TRPML), which have been shown to be required for NLRP3 inflammasome activation[47].

The $[Ca^{2+}]_{ex}$-induced IL-1β release of peripheral blood monocytes is significantly higher in RA patients than in healthy donors. While patients with PsA and SLE do not show an increased $[Ca^{2+}]_{ex}$-induced monocyte response, it is possible that chronic inflammation at sites of high $[Ca^{2+}]_{ex}$ concentrations triggers the same mechanism also in other diseases.

In arthritis, however, the described mechanism appears to contribute to joint inflammation, because increased $[Ca^{2+}]_{ex}$ concentrations are present at sites of bone erosions (for schematic representation of the pathway see Fig. 8j). Systemic hypercalcemia has also been reported to be associated with RA and linked to high disease activity[48], although this is not generally believed to be a key feature of RA. Anecdotal evidence indicates, that RA can lead to pronounced hypercalcemia if larger joints are involved[49].

In RA, bones of fingers and toes are often affected by periarticular demineralization, which occurs early in the disease before the erosive bone lesions develop. Demineralization predicts later joint damage, is associated with inflammation, and can be prevented by efficient therapy with tumor necrosis factor (TNF) inhibitors[50–53]. We propose that $[Ca^{2+}]$ efflux out of extracellular reservoirs of the bone in combination with generalized osteoporosis, which is also typical for the disease, leads to temporary local $[Ca^{2+}]_{ex}$ increases. At sites of osteoclastic bone resorption, fetuin-A is also available due to its abundance in mineralized bone[31], and resorption of bone matrix is linked to release of $[P_i]_{ex}$ as well[29]. Consequently, local $[Ca^{2+}]_{ex}$ increases accompanied by CPP formation are likely to occur and trigger NLRP3 activation, which in turn perpetuates and aggravates joint inflammation. The resulting IL-1β release is also a pivotal factor for bone erosion in RA, since IL-1β knock-out completely protects TNF transgenic mice from the severe erosive disease typically present in this model[54]. In addition, IL-1β positive, pro-inflammatory monocytes have recently been confirmed to represent a cell state expanded in RA[55].

Increased expression of the CaSR in RA monocytes could be caused by the systemic inflammation in those patients, but the lack of a correlation between disease activity and monocytic $[Ca^{2+}]_{ex}$-induced IL-1β release argues against that. Instead, it is feasible that monocyte precursor cells in the bone marrow in RA are already exposed to increased calcium concentrations in the vicinity of arthritic joints, leading to increased receptor expression.

In affected joints of RA, $[Ca^{2+}]_{ex}$ also determines the activity of the extracellular $Ca^{2+}$-dependent enzyme peptidylarginine deiminase 4 (PAD-4)[56], which citrullinates proteins in the joint and thereby triggers an autoimmune response in the form of anti-citrullinated protein antibody (ACPA) production. Local $[Ca^{2+}]_{ex}$ increases might therefore contribute to citrullination in the joints in RA, but also in coronary arteries, since published data showed that both citrullinated fibrinogen and vimentin were correlated with the coronary artery calcium score in RA patients[57]. This mechanism might even contribute to the development of coronary heart disease (CHD) in general, since citrullinated proteins and PAD-4 enzyme can also be detected within atherosclerotic plaques obtained from non-RA patients[58], while ACPA positivity also predicts the CHD risk in individuals not suffering from RA[59].

The pro-inflammatory effect mediated by CaSR stimulation and subsequent CPP-driven inflammation might explain the negative outcome of the EVOLVE trial[60], in which CKD patients were treated with Cinacalcet. This allosteric modulator of the CaSR enhances CaSR signaling, and significantly decreases serum $[Ca^{2+}]$ and parathyroid hormone levels, but failed to improve cardiovascular outcome, possibly due to the resulting CaSR-mediated, NLRP3-dependent IL-1β production triggered by Cinacalcet itself.

In summary, the intricate interplay between $[Ca^{2+}]_{ex}$, $[P_i]_{ex}$, and fetuin-A resulting in the formation of CPPs and the regulation of their uptake into monocytes/macrophages by CaSR suggests the existence of a homeostatic system that responds to and corrects excessive ion concentrations. Overload of this system is associated with pro-inflammatory responses from monocytes and macrophages including release of exorbitant IL-1β amounts, and could potentially be triggered by bone erosion, increased cell death or phosphate-rich diet.

Whether the $[Ca^{2+}]_{ex}$ and $[P_i]_{ex}$-mediated pro-inflammatory response is beneficial in case of perturbation of the homeostatic system, or whether additional harm and tissue destruction results from it, is likely dependent on the specific scenario present. In RA, the resulting monocyte activation followed by IL-1β release are detrimental due to the ensuing joint destruction.

## Methods

**Healthy donors and patients with RA.** Experiments with human monocytes were conducted in accordance with the Declaration of Helsinki according to a protocol approved by the Ethics Review Board of the Medical Faculty, Leipzig University (313/14-ek and 430/16-ek), with written informed consent from all blood donors. Age and sex matched healthy donors were recruited among healthy blood donors. 97 patients with RA (69 females and 28 males, with average age of 62 years) according to the criteria of the American College of Rheumatology were included in the study.

Monocytes were isolated from 75 RA patients to perform various experiments. Synovial fluid was obtained from 22 RA patients (14 females and 8 males, with average age of 55 years, TNF inhibitors ($n = 5$), conventional synthetic DMARDS ($n = 17$)) and 13 patients with other non-destructive arthritis (5× reactive arthritis, 6× osteoarthritis, 2× ankylosing spondylitis) by aspiration for therapeutic purpose. Synovial biopsy specimens were obtained from six patients with RA who underwent synovectomy at the Department of Orthopedic, Trauma and Plastic Surgery, Leipzig University (three females and three males, with average age of 51 years, conventional synthetic DMARDS ($n = 6$)). The study was approved by the Ethics Review Board of the Medical Faculty, Leipzig University (093-2008, 281-13-07102013), with written informed consent from all blood donors. The 39 patients with DMARD treatment included in the clinical analysis of $[Ca^{2+}]_{ex}$-induced IL-1β release received the following medications: methotrexate ($n = 22$), TNF inhibitors ($n = 13$: $n = 7$ Enbrel, $n = 4$ Adalimumab, $n = 2$ Certolizumab pegol), rituximab ($n = 3$) or the JAK inhibitor baricitinib in one patient.

**Monocyte isolation.** Peripheral blood mononuclear cells (PBMCs) were freshly isolated from human peripheral blood by density gradient centrifugation using Ficoll paque (GE healthcare). Monocytes were isolated from PBMCs by negative separation with the monocyte isolation kit II (Miltenyi Biotech) according to the manufacturer's instructions. Monocytes were cultured in macrophages differentiation media (RPMI1640 cell culture medium (Gibco, Lifetechnologies) with 2% AB serum, 1% penicillin–streptomycin (Invivogen), 50 mM 2-Mercaptoethanol (Gibco, Lifetechnologies), 1 mM sodium pyruvate (Gibco, Lifetechnologies), 1 mg/ml NaHCO₃ (Roth), 0.1% non-essential amino acids (Gibco, Lifetechnologies), 0.4% MEM vitamins (Gibco, Lifetechnologies)) for 7 days to differentiate them into macrophages.

**Cell culture and stimulation.** Monocytes were directly stimulated after isolation in either standard RPMI1640 cell culture medium (high $[P_i]$, 5.6 mM Na₂HPO₄, Gibco, Lifetechnologies) or customized RPMI1640 containing 1 mM Na₂HPO₄ (low $[P_i]$) supplemented with 10% FBS (Gibco, Lifetechnologies). $3 \times 10^5$ monocytes were seeded in 96-well plates and 100 ng/ml LPS (Invivogen) was used for monocyte "priming". Inhibitors were pre-incubated for 30–60 min prior to stimulation.

CaSR (established in our group) and NLRP3-deficient (Invivogen) THP-1 cells were cultured in RPMI1640/10% FBS/1% penicillin and streptomycin (pen/strep) and selection antibiotics as indicated in the data sheet. Assays were performed in 24-well plates. $5 \times 10^5$ cells/well were plated for differentiation in 50 ng/ml PMA

(Tocris)-containing medium for 2 days before LPS-priming (100 ng/ml) and stimulation.

The following reagents and inhibitors were used for several cell culture experiments. $Na_2HPO_4$, $BaCl_2$, fetuin-A from FBS were purchased from Merck, $CaCl_2$, sodium phosphonoformate tribasic hexahydrate, from Sigma, YM254890 from Wako chemicals, Calhex231, NPS2143, Latrunculin A, Cytochalasin D from Tocris, Latrunculin B, PAF C-16, LLOMe from CaymanChemicals, $MgSO_4$ from AppliChem, ATP from Roche, DMSO from Serva, N-fMLP from abcam, and CA-074-Me from Selleck-Chem.

### Crispr/Cas9 knockout of CaSR in THP-1 cells

THP-1 cells were transduced with lentiviral Cas9 particles (Dharmacon, Edit-R Lentiviral hEF1α-Blast-Cas9 Nuclease Particles, Cat. no. VCAS10126). $5 \times 10^4$ cells were transduced at a MOI 3 in 250 μl RPMI with 1% FBS without antibiotics. After 16 h transduction, medium was replaced by 500 μl RPMI1640/10% FBS. For selection, 10 μg/ml blasticidin was added to culture medium 48 h after transduction. The cells were selected for 1 week before further use. Blasticidin and puromycin (Invitrogen) concentrations for selection were established in advance by an antibiotic kill-curve.

Cas9 protein expression was confirmed by Western Blot (Cell Signaling, Cas9 antibody 7A9-3A3, Cat. no. 14697S). Cas9-transduced THP-1 cells were then transduced a second time, as described above, with lentiviral particles containing the sgRNAs against CaSR (Dharmacon, Edit-R Lentiviral mCMV-Puro-sgRNA Particles, clone VSGHSM_27523470, DNA target sequence 5′-GGACCTTCTTCAGGAATTCC-3′, Cat. no. VSGH10142) or a non-targeting control sequence (Edit-R Lentiviral mCMV-Puro non-targeting sgRNA particles, Cat. no. VSGC10216). After 48 h, transduced cells were selected by 0.8 μg/ml puromycin for 1 week. Further cell culture was the same as for THP-1 wildtype cells except for the addition of the two selection antibiotics.

To ensure that the transduced cells were not shedding virus particles, a p24 ELISA (Sino Biologicals, Cat. no. KIT11695) was carried out according to the manufacturer's instructions.

Cells were plated as single cell clones in a 96-well U-bottom plate on a BD FACS Aria III cell sorter (Core Unit fluorescence technology, Leipzig University with the help of Kathrin Jäger). For better growth rates conditioned medium from ongoing THP-1 cell culture was mixed 1:1 with RPMI1640/10% FBS, 1% pen/strep but without selection antibiotics.

After 3 weeks clones were screened in a Mismatch-Detection Assay (Takara, Guide-it Mutation Detection Kit, Cat. no. 631443). Sequences of the primers for mismatch detection in the CASR gene were designed spanning the sgRNA target site (forward 5′-TGCAGCTGATGACGACTATG-3′ and reverse 5′-CTAAACCTGTCGCCACTTTCT-3′). One clone positive in the mismatch assay at the CaSR site (CaSR70 B6) and one non-targeting sgRNA clone not containing a mismatch (Ctrl216 B6) were then used for further experiments. See supplementary Fig. 3e for mismatch assay results. Deletion of base pairs at the target sequence was verified by Nextera DNA library preparation (Illumina) and sequencing of PCR products in the DNA core unit (Leipzig University). All sequences of CaSR70 B6 contained 13-bp, 17-bp, or 18-bp deletions, which ruled out persistence of the wild-type gene, and were not present in Ctrl216 B6 cells (see Supplementary Fig. 3f).

### Mice

Experiments were performed using male/female C57BL/6 mice (wild-type and CaSR mutants). Mice were bred and maintained under specific pathogen-free conditions (ambient temperature $22 \pm 2$ °C, humidity $55 \pm 15$%, and 12 h dark/light cycle) at the animal facilities at Medizinisch Experimentelles Zentrum, University of Leipzig, Germany. B6.129P2-Lyz2tm1(cre)Ifo mice (LysM-Cre) were purchased from The Jackson Laboratory, and CaSRflox/flox mice were kindly provided by Wenhan Chang[61]. CaSRflox/flox mice were bred with transgenic mice expressing Cre Recombinase under the control of the LysM promoter and genotyped prior to all experiments. Mice were used at 2–8 months of age.

All animal experiments were approved by the local Animal Care and Use Committees of the State of Saxony, Germany, as recommended by the Animal Ethics Review Board (Regional Administrative Authority Leipzig, Germany, T29/14, TVV27/19) and followed the NIH guidelines for care and use of animals.

### Experimental arthritis

CIA was induced in 10 DBA/1J mice (Harlan Winkelmann) at 7–8 weeks of age by immunization with 50 μl of a 1:1 (v/v) emulsion of CFA and 0.1 M acetic acid containing 50 μg of chick type II collagen (CII; Chondrex) and 50 μg of heat-killed Mycobacterium tuberculosis (Chondrex) at the base of the tail. The clinical severity of arthritis was quantified as follows: 0: no joint swelling, 1: swelling of one finger joint, 2: at least two swollen finger joints, 3: mild swelling of wrist or ankle, and 4: severe swelling of wrist and ankle. Scores of all forepaws and hind paws were totaled for each mouse. Blood sampling by heart puncture, peritoneal lavage, bone marrow flushes from the cavities of femur bones, and clinical scoring of arthritis severity were all performed on day 40 post injection, when the mice were sacrificed.

To test R568, CAIA was induced in 15 DBA/1J mice by injection of 0.5 mg antibody cocktail (ModiQuest Research, Netherlands) and 10 μg LPS (Sigma) i.p. 5 mice were additionally daily injected with 150 μl of 33 μM R568 i.p., and 10 mice with the control solution sodium acetate. Arthritis severity was evaluated daily.

Blood was collected by heart puncture, mononuclear cells were isolated by Ficoll-Paque™ centrifugation, and CD11b+ monocytes were isolated by positive magnetic bead separation (Miltenyi Biotech). IL-1β in cell culture supernatants was measured with the mouse IL-1β ELISA (BD Bioscience).

### Measurement of human IL-1β

Human IL-1β ELISA Set II (BD Bioscience) was used to detect secreted IL-1β in cell culture supernatants after stimulation of monocytes for 16 h or THP-1 cells for 8 h at cell culture conditions (37 °C/5% $CO_2$). ELISA was performed as described in manufacturer's instructions.

### Staining of ASC-specks

To visualize inflammasome activation, ASC-speck formation was detected by immunological staining of ASC. $3 \times 10^5$ monocytes were seeded in 96-well cell imaging plates (MoBiTec) and stimulated for 8 h prior fixation with 4% paraformaldehyde (PFA, Merck) for 20 min at room temperature. After washing with PBS, unspecific antibody binding was blocked with PBS/50% human AB-serum for 30 min. Subsequently, anti-ASC antibody (1:500 in PBS/10% AB-serum, polyclonal rabbit anti-ASC (AL177, AdipoGen Life Science)) was incubated for 30 min followed by washing with PBS and anti-rabbit-AlexaFluor488 (1:100, Biolegend) incubation for 30 min. After washing with PBS, cells were imaged with a fluorescence microscope (Zeiss Axio Observer.Z1/AxioCamMRm3) and processed with Zeiss ZEN 2 software.

### DMR measurement

DMR was analyzed with the Corning® EPIC® Biosensor system and Epic Quest (R) 2.1.0.2. Analysis of monocytes was directly done after their isolation from peripheral blood. $6 \times 10^4$ monocytes per well were seeded in non-coated EPIC® microplates in RPMI1640 cell culture medium supplemented with 20 mM HEPES. Before stimulation, monocytes were pre-incubated with 100 ng/ml LPS ± inhibitor for attachment to the bottom of the plate (60 min). Monocytes were stimulated with compound solution and incubated in a total volume of 40 μl for up to 4 h at 28 °C during detection of DMR. To calculate stimulus-dependent DMR, values of unstimulated (LPS ± inhibitor) monocytes were subtracted from DMR response values of stimulated (LPS ± inhibitor ± stimulus) monocytes. The same protocol was performed for THP-1 cells, which were seeded for differentiation at a density of $2.5 \times 10^4$ c/w in non-coated 384-well assay plates in medium containing 50 ng/ml PMA 2 days before the experiment.

### CPP preparation and stimulation

For preparation of CPPs 200 μl RPMI1640/10% FBS (room temperature) were mixed with 5 μl 100 mM $CaCl_2$ (2.5 mM $[Ca^{2+}]$) in a 1.5 ml Eppendorf tube, vortexed and centrifuged for 2 h at $16,000 \times g$ at 21 °C. For calcein-stained CPPs, "CPP-medium" was supplemented with 15 μM calcein (Sigma). After pelleting of spontaneously formed CPPs for 2 h, supernatant was carefully removed. For stimulation with CPPs, the CPP pellet was resuspended with 1 mM $[P_i]$ containing cell culture medium to prevent the formation of new CPPs. 1x CPPs describes the CPP content out of 200 μl "CPP-medium" from one 1.5 ml tube, 2x CPPs describes the CPP amount out of two tubes. The 2x CPP concentration was used for stimulation, if not indicated otherwise. For detection of fetuin-A in CPPs, the CPP pellet (3x CPP) was washed in FBS-free 1 mM $[P_i]$ RPMI1640 and again centrifuged for 2 h before resuspension in Laemmli buffer (reducing conditions), and loading onto a 10% acrylamide-SDS-gel and transfer to a PVDF membrane (GE Healthcare) via wetblot. Fetuin-A was detected with the goat-anti-Fetuin-A antibody (N-20, Santa Cruz). CPP medium was filtrated with a Whatman® Anotop® sterile syringe filter 0.1 μm.

### CPP analysis via dynamic light scattering

DLS was performed on a Zetasizer Nano ZS (Malvern Instruments, Herrenberg) with a He/Ne laser operating at a wavelength of $\lambda = 633$ nm utilizing a detection angle 173° backscatter. Three measurements (a 3–20 runs) were carried out at 25 °C after equilibration. Mean main peak intensity or Z-average was used for approximation of particle size. Data were analyzed with Malvern Zetasizer Software 7.11. Measurements were supported by the group of Stephanie Hoeppener, Jena Center for Soft Matter (JCSM) and the group of Lukas Wick (Birgit Würz), Helmholtz Center for Environmental Research—UFZ Leipzig.

### CPP analysis and detection of CPP uptake with TEM-EDX

TEM images were acquired with a FEI Tecnai G² 20 transmission electron microscope (FEI/Thermo Fisher) operated at an acceleration voltage of 200 kV. Most frequently images were recorded on an Olympus Soft Imaging Solution (OSIS) Megaview (1k) or an Eagle 4k HS CCD camera system. Imaging processing was performed utilizing ImageJ 1.52 or Fiji. Particle characterization was performed utilizing cryo-TEM or utilizing sample blotting on carbon-coated TEM grids (Quantifoil, Germany), respectively.

### Cryo-TEM investigations

For these studies, 8.5 μl of the aqueous water solution containing CPPs were blotted onto Quantifoil R2/2 grids (Quantifoil, Germany) by means of a Vitrobot Mark IV. Samples were vitrified utilizing liquid ethane. After blotting and plunge freezing samples were transferred into a Gatan cryo-stage and maintained at liquid Nitrogen temperature until transferred to the TEM utilizing a Gatan 626 cryo-holder.

**Particle size analysis**. Fifteen microliters of the solutions were blotted onto carbon-coated TEM grids and analyzed by TEM investigation. Initially a comparative study was performed to check if drying the particles on the TEM-grid does alter their shape or size. This was not the case and, hence, all studies on the particle size evolution, supplementing the DLS studies, were performed utilizing this approach. Particle size analysis was performed by Origin 9.0 with data extracted manually from TEM images.

**EDX analysis**. EDX spectra were acquired with a Bruker Quantax system. Qualitative mapping results are presented as a color-coded overlay of preselected elements. A low background FEI EDX holder was used to minimize background signals. Naturally, the Cu originating from the utilized TEM grids and minor signals from the chamber could not be avoided. Selection of the elements was based on an overview scan with integrated element analysis in spectral mode. Attempts to access the quantitative composition of the particles showed different results and were not further quantified in the present study. Spectra obtained on the individual particles were acquired by integrative scanning of the beam in STEM mode over the respective particle and integration of the obtained signals.

**Cell stimulation and preparation for TEM investigations**. $5 \times 10^6$ freshly isolated peripheral blood monocytes were stimulated with 100 ng/ml LPS and 2.5 mM $CaCl_2$ in six-well plates coated with 1.5% RPMI1640-agarose (Serva Electrophoresis) in RPMI1640/10% FBS cell culture medium. After incubation at 37 °C/5% $CO_2$ cells were directly fixated in cell culture medium with 2.5% glutaraldehyde and 4% PFA (Electron Microscopy Sciences) for 1 h at room temperature followed by washing with PBS.

Ultrathin slices of the incubated cells were obtained by post-fixation of the fixed cells with osmium tetroxide (Science Services) for 1 h at 4 °C. Subsequently, the sample was dehydrated in a graded series of ethanol (Acros Organics, 30%, 50%, 60%, 70%, 80%, 90%, 100%). The obtained cell pellet was transferred into beam capsules and pre-embedded with a 1:2 mixture of Embed 812 epoxy resin (Science Services) and ethanol for 1 h, followed by incubation with a 1:1 mixture for 1 h and subsequent immersion of the pellet in pure Embed 812 resin (Science Services) overnight. Finally, resin and DMP were mixed to obtain a curable resin. The pellet was incubated with the activated resin mixture and cured for >24 h in an oven at a temperature of 80 °C.

The obtained sample blocks were trimmed to a size of ~1 × 1 mm and sliced with an RMC Powertome PTX (RMC, Boeckler) to obtain ultrathin sections with a thickness of ~80 nm. Slices were floated onto carbon-coated TEM grids (Quantifoil). Post-staining was occasionally used to improve the contrast but was avoided in cases where EDX analysis was carried out to minimize the presence of signals originating from the stains. Imaging of the slices was performed either in bright field-mode or by utilizing STEM to improve the contrast.

**Statistics and reproducibility of EM experiments**. Particles have been characterized by cryo TEM (Fig. 1h, n = 2) to determine the characteristic shape of the CPP particles after prolonged incubation times of several days. Further characterization studies were performed on samples blotted onto hydrophilized carbon-coated support grids, taking into account the peculiar structure of the CPPs which allows their unambiguous identification from precipitates and deposits of the buffer solution (RPMI1640/10% FBS and 2.5 mM $[Ca^{2+}]$). Particle size investigations of the CPPs was conducted after 2 h incubation time and is depicted in Fig. 1g. 230 independent CPPs were analyzed for the size determination (n = 1).

EDX investigations were performed on two arbitrary selected areas containing several CPP nanoparticles, which were blotted after long incubation times of several days (example shown in Fig. 1i). Additionally, >5 spectra of additional individual CPPs were acquired (example see Fig. 1j) per sample.

Three independently prepared samples were investigated to study the CPP uptake in human monocytes. Uptake was confirmed in two of the samples stimulated with CPPs. At least two sample areas showing the uptake events were selected and further investigated by EDX mapping (Fig. 1n). Additionally, five to six individual CPPs were further analyzed by acquiring the full EDX spectra (spectra for locations 1 and 4 are depicted in Fig. 1m, the EDX spectra on the other locations can be found in Fig. S2b) between 0 and 12 keV. Locations of the analyzed CPPs are marked in the STEM image depicted in Fig. 1l.

**Confocal Raman microspectroscopy**. Confocal Raman imaging was used for the detection of calcium-phosphate nanoparticle uptake in monocytes. Raman imaging was done with the help of Tom Venus in collaboration with the Institute of Medical Physics and Biophysics, Leipzig University. Monocytes were freshly isolated from human peripheral blood and $1 \times 10^6$ monocytes were seeded in Eppendorf cell Imaging dishes (145 μm glass bottom) in RPMI1640 medium to reach ultra-adherence for 60 min (37 °C/5% $CO_2$). Medium was replaced by 10% FBS containing RPMI1640. Monocytes were imaged before and after adding 2.5 mM $[Ca^{2+}]$ with a WiTec alpha300 R+ confocal Raman microscope with an excitation wavelength of 532 nm (34 mW, 50 μm pinhole) and a ×63/1 Zeiss W Plan Apochromat objective under temperature control (38 °C). Raman spectra were collected pixel-wise using a 600 g mM$^{-1}$ grating, pixel size of 250 μm × 250 μm and an

integration time of 70 ms. Images were taken and processed with the WiTec software Control FOUR and Project FOUR PLUS software.

**$[Ca^{2+}]$ measurement in cell culture medium**. $Ca^{2+}$ concentration was measured as described previously[2]. Addition of 0, 0.5, 1.0, 1.5, 2.0 or 2.5 mM $[Ca^{2+}]$ to RPMI1640/10% FBS resulted after 2 h in the following directly measured $[Ca^{2+}]$ values: 0.6, 0.9, 1.2, 1.5, 1.6, or 1.7 mM (for more details see ref.[2]).

Synovial fluid was obtained by anaerobic sampling and the $Ca^{2+}$ concentration was measured with the radiometer ABL 90 series (Radiometer GmbH).

To determine $[Ca^{2+}]$ in bone marrow of mice, bone marrow was obtained by flushing the femur once with 100 μl 0.9% NaCl. Cells were removed by centrifugation. $[Ca^{2+}]$ was measured in the supernatant. The final $[Ca^{2+}]$ was calculated by multiplying the measured $[Ca^{2+}]$ in 100 μl with the dilution factor (ratio between the 100 μl flush volume and the calculated volume of the flushed bone marrow cavity of the femur).

**Detection of CaSR protein expression by Western Blot**. Whole cell extracts from freshly isolated monocytes from six RA patients and seven healthy donors were obtained by lysing $2 \times 10^6$ cells in RIPA lysis buffer and incubated in non-reducing Laemmli buffer for 30 min at room temperature. Samples were resolved by SDS–PAGE and transferred to a polyvinylidene difluoride membrane (GE Healthcare) using a transfer apparatus according to the manufacturer's protocols (Bio-Rad). The membrane was incubated with Ponceau S solution (P7170, Sigma-Aldrich) for 60 min to check for equal loading and transfer of proteins. The membrane was blocked with 5% soy protein (Vitasyg) in TBST (10 mM Tris, pH 8.0, 150 mM NaCl, 0.5% Tween 20) for 60 min, then washed thrice, for 10 min each, with TBST and incubated with antibody against CaSR (Alomone labs, Cat# ACR-004) (1:200) overnight at 4 °C. Membrane was washed three times for 10 min and incubated with a 1:2500 dilution of anti-rabbit antibody (7074 S, Cell Signaling) in 5% soy protein in TBST for 1 h at room temperature. Blots were washed with TBST thrice and developed with the ECL system.

The intensity of the bands was measured using ImageJ 1.52 software and the ratio of intensity of the CaSR band to the intensity of Ponceau S staining of the respective lane was calculated.

**Immunochemistry**. Frozen synovial tissues embedded in OCT compound were cut into 5 μm sections, fixed in acetone for 10 min, and air-dried. After blocking with normal goat serum, slides were incubated with rabbit-polyclonal anti-CaSR Ab (Santa Cruz, clone H100) for 1 h. Following incubation with secondary goat-anti-rabbit Ab conjugated with either peroxidase or phosphatase, stains were developed with the substrates 3,3-diaminobenzidine tetrahydrochloride (DAB) or 3-amino-9-ethylcarbazole (AEC), respectively. Sections were counterstained with hematoxylin.

Ionized calcium in synovial tissue sections was stained with Calcium red (Glyoxal-bis(2-hydroxyanil, GBHA) for 5 min. In the indicated experiments, synovial tissue sections were pre-incubated with 20% EDTA solution.

**Detection of macropinocytosis**. Calcein-stained cell culture medium or calcein-stained CPPs were used to detect macropinocytosis of monocytes. For the latter, CPP formation was induced in the presence of 15 μM calcein. Freshly isolated monocytes were seeded on 1.5% RPMI1640-agarose (Serva) in customized or standard RPMI1640 cell culture medium in the presence of 15 μM calcein or 1x calcein-stained CPPs. Monocytes were stimulated for 45 min at 37 °C/5% $CO_2$ and subsequently washed twice with PBS/1% BSA or PBS only. Calcein uptake was detected by flow cytometry (LSR II, BD Biosciences, BD FACS Diva 8.0.6) or Amnis® ImageStream$^X$Mark II analysis (Ch04—Brightfield, Ch02—calcein, INSPIRE for the ISX mkII Version 200.1.388.0). Data were analyzed with FlowJo V10.1 software or IDEAS 6.2 and percentage of stimulus-dependent uptake was calculated by referring to LPS-triggered uptake as a control (5% of calcein-positive monocytes as threshold).

For analysis in WT and CaSR-deficient THP-1, cells were seeded on 1.5% RPMI1640-agarose in RPMI1640/1% penicillin and streptomycin (pen/strep) without FBS and incubated at 37 °C/5% $CO_2$ for 16 h. $3 \times 10^5$ cells/well (48-well plate) were then seeded on 1.5% RPMI1640/1 mM $[P_i]$-agarose in RPMI1640/1 mM $[P_i]$ and were stimulated with 2x calcein-stained CPPs and 2.5 mM $[Ca^{2+}]$ for 10, 20, 30, and 60 min. After washing with PBS/1% BSA for two times, cells were fixed with 4% PFA for 15 min at 4 °C. THP-1 cells were washed with PBS and CPP uptake was detected by Amnis® ImageStreamXMark II analysis. Data were analyzed with IDEAS 6.2 and percentage of stimulus-dependent uptake was calculated by referring to $[Ca^{2+}]$-independent uptake as a control.

**Detection of cathepsin activity**. Magic Red Cathepsin-B Assay (ImmunoChemistry Technologies) was used for the detection of Cathepsin B activity in monocytes. Assay was performed as described in manufacturer's instructions. Cathepsin activity was determined after 3 h of incubation in either 1 or 5.6 mM $[P_i]$-containing cell culture medium. Fluorescence was measured with a Tecan infinite M200 plate reader (Ex 592 nm/ Em 628 nm) and Tecan Magellan V7.2.

**Detection of lysosomal leakage**. Lysosomal leakage was detected as described previously[42]. In brief, monocytes were seeded on 1.5% RPMI1640-agarose and loaded with 2 µg/ml acridine orange (Thermo Fisher) for 20 min at 37 °C/5% $CO_2$. After washing, monocytes were resuspended in fresh cell culture medium and seeded in new agarose-coated plates. Monocytes were stimulated with 100 µg/ml MSU (Invivogen) as control. Further, 500 µM LLOMe (CaymanChemicals) (added 10 min before analysis) was used to induce lysosomal rupture and therefore to discriminate between ruptured/leaky lysosomes and functional lysosomes for analysis. Loss of acridine orange staining (Ex 488 nm/ Em 650 nm) was used as indicator for lysosomal leakage and was detected with Amnis® ImageStream$^X$Mark II analysis. Analysis was done with IDEAS 6.2 software.

**Detection of cell death**. To compare cell death of monocytes stimulated in low/ high $[P_i]$ medium PI (dead cells) and Hoechst 33342 (all cells) (Thermo Fisher) staining was used and viability was analyzed using the dead over total analysis of Celigo® S Imaging Cytometer (Nexcelom). Monocytes were stimulated for 16 h at 37 °C/5% $CO_2$ and were subsequently analyzed for their viability.

Pierce LDH Cytotoxicity Assay Kit (Thermo Scientific) was used to analyze cell death in THP-1 CaSR deficient and NLRP3-deficient cells. $5 \times 10^5$ c/w were seeded in RPMI1640/10% FBS/50 ng/ml PMA in 24-well plates for differentiation 2 days before stimulation. Maximal LDH release was detected by lysing cells before LDH measurement. Assay was performed as described in manufacturer's instructions.

**Graphs and statistics**. Boxes in box-whisker-plots indicate 25–75% percentile, while whiskers show min/max values and the vertical lines indicate the median. Bar charts represent mean + SEM. Values of each experiment are represented as symbols in bars or in box-whisker-plots. All graphs and statistics were prepared with GraphPad Prism 8.0.1. Statistical significance was determined using the two-tailed non-parametric, unpaired Mann–Whitney $U$ tests or paired Wilcoxon test or $t$-test for samples sizes below $n = 5$, confidence interval of 95%. $p$-Values are indicated as $*p < 0.05$; $**p < 0.01$; $***p < 0.001$, $****p < 0.0001$, or $^\#p < 0.05$; $^{\#\#}p < 0.01$; and $^{\#\#\#}p < 0.001$. If not indicated otherwise asterisks indicate the statistical comparison to control conditions.

**Reporting summary**. Further information on research design is available in the Nature Research Reporting Summary linked to this article.

## Data availability

The datasets generated during and/or analyzed during the current study are available from the corresponding author on reasonable request. In addition, raw data are available as a source data file. Source data are provided with this paper.

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

## Acknowledgements

We would like to thank Cornelia Arnold and Ute Posselt for technical support. This work was supported by the German Research Foundation (www.dfg.de/): U.W. (WA 2765/9-2), M.R. (RO 4037/3-1), C.S. (STA 1265/3-1), T.S., E.J. SFB 1052 (B6) (Project number 209933838), N.S. (STR 477/14-2), by the European Fonds for Regional Development (EFRE): (S.H., P.S.), by the European Social Fonds (https://www.esf.de/) (A.P.), by the CRC "PolyTarget" 1278 (316213987) by project C04/Z01 (P.S. and S.H.), and by research funding of the Medical Faculty, University Leipzig (https://www. uniklinikum-leipzig.de/wissenschaft-forschung/forschungs-administration/forschungsförderung) (C.S.). The funders had no role in study design, data collection and analysis, decision to publish, or preparation of the manuscript. Open access funding provided by Projekt DEAL.

## Author contributions

Experiments and data analysis: E.J., S.M., C.S., M.H., M.R., S.R., V.R., K.R., S.S., M.G., S.J., M.P., P.S., S.H., A.P., C.S., N.R., I.E.-L., T.V., K.K. Resources: S.J., S.W., M.H., R.S., O.S., M.P., W.C. Visualization: E.J., C.S., S.M., M.R., S.H., T.V. Project administration: U.W., M.R. Writing—original draft: U.W., E.J., S.M., M.R. Writing—review & editing: T.S., E.J., S.M., C.S., M.R., A.P., C.S., I.E.-L., N.S., S.H., U.W. All authors have approved the final version of the manuscript.

## Competing interests

The authors declare no competing interests.
