## [Peer Review File · Nature Communications]

Reviewers' comments:

Reviewer #1 (Remarks to the Author):

In the current manuscript, Jager et al. suggested a mechanism how CaSR signal activates the NLRP3 inflammasome. Previously, it has been reported that activation of CaSR by extracellular calcium or allosteric agonists induce NLRP3 inflammasome activation through intracellular calcium influx by the same group of this manuscript and others. Here, authors claim a new mechanism. Extracellular calcium forms calciprotein particles (CPP) in concert with phosphates and calcium-binding serum protein, fetuin-A that is present in FBS of cell culture media. Activation of CaSR induces internalization of CPP into LPS-primed monocytes through micropinocytosis, which activates the NLRP3 inflammasome. Further, authors provided *in vivo* evidence of the CaSR-CPP-NLRP3 inflammasome connection from RA patients' monocytes and RA mouse models, in which monocytes showed heightened releases of pro-inflammatory cytokines including IL-1 β through increased calcium concentration in synovium and CaSR expression. The authors provided a lot of data to prove the result. However, there are major points in some of data to be addressed in prior to drive conclusion from the results. And some of data are missing some important control experiments.

1. In Fig. 1a, CaSR-mediated inflammasome activation by extracellular calcium was probed using CaSR-deficient THP-1 cells. Although the released IL-1 β from CaSR-deficient cells were lower than WT THP-1 cells for each treatment of calcium concentration, CaSR-deficient cells still released substantial amount of IL-1 β in a dose-dependent manner. Curiously, ATP or MSU, very well-known NLRP3 activators, induced IL-1 β release not that much and even much lower than the level of IL-1 β released from CaSR-deficient cells stimulated by calcium. This result may suggest another pathway of calcium-induced inflammasome activation in addition to CaSR-mediated NLRP3 activation. To address this question, it is suggested to include data from NLRP3-deficient THP-1 cells that was used in Fig. 9l of this study.

2. In the current study, most of calcium-induced inflammasome activation experiments were done by 16h treatment of LPS-primed monocytes. Usually, IL-1 β release from monocyte/macrophages by inflammasomes is happened in two steps, priming and activation. After priming by such as LPS, the activation step occurs within relatively short period (1-3h) except crystal/particulate-induced NLRP3 activation, which is needed 6-16h (Martinon et al., Gout-associated uric acid crystals activate the NALP3 inflammasome. 2006, *Nature*, 440:237-241). Thus, it is possible that calcium-CPP-mediated NLRP3 activation is not only CaSR-mediated but also crystal/particulate-induced by CPP. Actually, authors showed that calcium-induced CPP uptake was happened within 30 min (Fig. 2k). Since engulf of CPP is the important step for CaSR-mediated NLRP3 activation, the NLRP3 inflammasome is expected to be activated within few hours. Thus, author should provide data that can exclude this possibility by showing early time points and also by including additional data for both MSU and ATP stimulations as controls in several data, particularly Fig.3b (need MSU); Fig.3c (need ATP); Fig.3f (need MSU); Fig.3g (need both ATP and MSU); Fig.4a (need both ATP and MSU); Fig. 4i (need MSU).

3. In Fig 5, for the data from CAIA model and CIA model, it is suggested to include data for treatment with CaSR antagonists, which are expected to attenuate RA symptoms.

Reviewer #2 (Remarks to the Author):

Jager and colleagues study the role of calcium in the activation of NLRP3 inflammasome complexes with relevance for arthritis. Previous studies linking calcium to NLRP3 raised queries, and speculation that in fact, this was an artefact of the system due to calcium precipitation into NLRP3 activating particles (Muñoz-Planillo et al *Immunity* 2013). However data implicating a role for the

CaSR suggested a more direct signalling role, and this current work helps to now explain that, because it is required for macropinocytosis of the Calcium containing particles. Therefore, this manuscript advances the field to link calcium to NLRP3 in the context of arthritis, and explains divergent results at the same time.

Major points

It is particularly appealing that CaSR is identified as a potential target preventing macropinocytosis of calcium containing particles and NLRP3 activation. Other particles, such as MSU, also trigger the inflammasome, and deletion of NLRP3 or downstream pathway members ASC or GSDMD can temporarily restrict IL-1 β production and cell death in vitro, however alternate mechanism exist to eventually trigger cell death (Rashidi et al JI 2019). Ultimately, this means that in vivo, targeting NLRP3/ASC/GSDMD had little physiological effect on MSU induced inflammation. Therefore, CaSR could potentially prevent calcium particle uptake, and all subsequent forms of downstream cell death. The authors could potentially perform some experiments to confirm this, and compare against other NLRP3 activators.

Reviewer #3 (Remarks to the Author):

This is a very interesting study that links the formation of calcium-, phosphate- and fetuin-containing nanoparticles (CPPs) to inflammatory release of IL-1 β . Moreover, the authors provide a rationale for the increased inflammation observed in rheumatoid arthritis, which is suggested to be due to increased CPP uptake and increased calcium-sensing receptor (CaSR) expression.

The data are interesting and convincing. Some of what's presented is already established (e.g. the formation of CPPs by critical calcium and phosphate levels, inclusion of fetuin, uptake of CPPs by monocytes and macrophage, CPP-stimulated IL-1 β release etc). The data that speaks to established knowledge enlarges the manuscript somewhat, but is necessary to complete the narrative in this study. Overall, this is a significant advance.

A fundamental missing point is the link between increased CPP uptake, enhanced lysosomal calcium content and NLRP3 inflammasome activation. The authors speculate that cytosolic calcium signals arising via TRPML channels are involved, and illustrate that in their final model diagram, but do not explore that aspect.

The authors should explicitly comment that they evidently have both CaSR-dependent and CaSR-independent IL-1 β release in their system. The knockout of CaSR reduced, but did not abrogate, the effects they report.

CPPs were taken up by monocytes and some IL-1 β was produced under near-physiological conditions (e.g. 1 - 2 mM Ca²⁺ext), in some of the data shown. Where is the threshold for pathological inflammation?

A few minor points that need attention:

1. There are a few typographical errors. For example, page 4, line 24. 'adenosin' should be 'adenosine'.
2. Legend to Figure 1. '[Ca²⁺]' needs a superscript '2'.
3. Figure 1a. Given the size of the error bars on the graph, the response to ATP doesn't look to be that significant compared to the zero calcium control. Please confirm there was a significant response to ATP. Why are the responses to ATP so different in Figures 1 and 3?
4. Page 6, line 19. '... over-representation of calcium...' doesn't really make sense ('over-representation' suggests error in measurement or presentation of data). I presume the authors mean an excess of calcium over phosphorus after 8 hours?
5. The labelling of the y-axis in Figure 2 is non-conventional and misleading. 'Calcein uptake in %' should be replaced with 'Cells with calcein fluorescence (%)', as that is what the data show. Please

try to use a similar labelling format (Figure 2i) has a slightly different y-axis label style.

6. Did the calcein enter the cells by macropinocytosis from free solution, or did the calcein bind to CPPs that were then taken into cells? Either way, the increased micropinocytosis due to Ca^{2+} -ext is clear, but I am wondering if the calcein uptake is proportional to CPP uptake in the assay shown.

7. Page 8, line 8. '... und...' should be '... and...'.

8. Figure 2i. The calcein-labelled particles with the monocytes appear to be much larger than the 60 – 100 nm particles observed in EM (even accounting for light scattering). Please explain the variations in size of the intracellular particles.

9. Please explain what the DMR measurements are indicative of. An inexperienced reader needs to understand what is being measured.

10. The variation in amount of IL-1 β measured in different experiments is surprising. There is a 10-fold difference in the y-axis scales from different experiments (even with strong stimulation conditions). What's the basis of the variation?

Reviewer #4 (Remarks to the Author):

I have been asked to specifically comment on the particle characterization. Overall the TEM and EDX spectroscopy presented are good, and I have a few small corrections / questions:

-The EDX spectra presented should have all elements identified, not only those that are being discussed (even if due to the holder/microscope, sample preparation and support grids this can be – and in part already is – explained). This specifically relates to Figure 1J and 1M, and supplementary Figure 2B.

-Does the EDX spectrum in Figure 1J relate to that in Figure 1I, if so this should be stated.

-Figure 1K is described in the caption as a TEM image, however the contrast suggests this is a HAADF STEM image and as such should be corrected.

-The figure caption for Figure 1M is confusing – what is the '(d + e)' referring to?

-In the methods section on 'Particle size analysis' the statement 'Initially a comparative study was performed to check if imaging artifacts due to the drying process were observed.'. Is the sizing study here for primary particle size measurements? If so, this should be stated. While primary particle size measurements are readily achieved by TEM, specialist approaches are required to prevent artefacts thereby allowing analysis of any agglomeration of particles (which I do not believe has been attempted here).

-In the methods section on 'Cell stimulation and preparation for TEM investigations' the sample block are presumably 1 by 1 mm (rather than mM).

Reviewer #5 (Remarks to the Author):

The authors propose that calcium is a major mediator of inflammation through activation of the inflammasome in RA. While the hypothesis is interesting and some of the data are compelling, a number of issues should be addressed.

One overarching concern is the specificity for RA and whether this is a cause or effect of inflammation. The authors argued first that RA monocytes are more sensitive to the calcium and then later state that the concentration of calcium is higher in RA and is therefore responsible for the biological effects observed. In addition, the authors show that IL-1 production is increased in

CIA monocyte/macrophages compared to non-arthritic mice. Thus, the increased sensitivity to calcium could be a response to inflammation rather than a central abnormality of RA pathogenesis. Additional controls in monocytes from patients with other forms of chronic arthritis are needed to determine whether this is an RA phenomenon or a broader response inflammation. An additional pre-clinical models would be important to determine whether this phenomenon is specific to synovial inflammation.

1. Many of the drugs used in RA are known to affect cytokine production, including prednisone and methotrexate. In most cases, they would tend to decrease cytokine production rather than increase production. Nevertheless, the authors should state what medications the RA patients were taking and determine whether cytokine release results correlated with IL-1 production?
2. Figure 4e shows bell shaped dose response, which is always a concern in biological models. Because the authors are attempting to mimic the effects of SF, they should directly assess the effect of SF on monocytes and whether calcium levels in SF correlate with cytokine release.
3. Figure 5c—Quantification and statistical significance is needed
4. Figure 5d—The differences are statistically significant, but quite small. The authors evaluate the model more completely, including effects on histology, bone damage, and synovial cytokine expression to determine whether the mechanism is related to the in vitro effects of calcium on monocytes.
5. Figure 5f—The authors state that calcium and IL-1 are “closely correlated”. The results are statistically significant, but are not particularly striking in terms of the R value.
6. Figure 5i—Insufficient information is provided on the protocol for CIA or the time points where cells were harvested.
7. Figure 5j—Not necessary and could be deleted
8. Discussion—Although there is a 2006 paper suggesting that hypercalcemia is correlated with RA severity, abnormalities of calcium levels are not consistently recognized as a key feature of RA.

Response to the Referees

We would like to thank all referees very much for the constructive criticism and the suggested experimental strategies. We have added several important experiments, and parts of the manuscript were rewritten in response to the reviewers comments. We believe, that the manuscript has greatly benefited by those suggestions. In the following, we will address the points raised by each referee separately.

Reviewer #1:

1. In Fig. 1a, CaSR-mediated inflammasome activation by extracellular calcium was probed using CaSR-deficient THP-1 cells. Although the released IL-1beta from CaSR-deficient cells were lower than WT THP-1 cells for each treatment of calcium concentration, CaSR-deficient cells still released substantial amount of IL-1beta in a dose-dependent manner. Curiously, ATP or MSU, very well-known NLRP3 activators, induced IL-1beta release not that much and even much lower than the level of IL-1beta released from CaSR-deficient cells stimulated by calcium. This result may suggest another pathway of calcium-induced inflammasome activation in addition to CaSR-mediated NLRP3 activation. To address this question, it is suggested to include data from NLRP3-deficient THP-1 cells that was used in Fig. 9I of this study.

Response: We thank the reviewer for this comment, since we agree that the strong calcium-induced IL-1 β production even in calcium sensing receptor deficient cells, which exceeds the levels achieved by the positive controls ATP and MSU (shown in Figure 1a), is unexplained. To corroborate this result, we repeated the experiments with calcium sensing-receptor-deficient peripheral blood monocytes of ^{Lyz2tm1(cre)llo}/CaSR $\Delta^{flox/\Delta^{flox}}$ mice in comparison to an MSU control. The results were comparable to calcium sensing-receptor deficient THP-1 cells, indicating that an additional, as yet unidentified mechanism must also contribute to calcium-induced IL-1 β production. In addition, these results confirmed that Ca²⁺ can be a much stronger inflammasome activator compared to the well-known NLRP3 activator MSU crystals.

As suggested by the reviewer, we confirmed the role of NLRP3 in this process using NLRP3 deficient THP1 cells. The results shown in supplementary figure S1a confirm, that at least 80% of the released IL-1 β concentrations are strictly dependent on NLRP3. As an additional experiment with freshly isolated peripheral blood monocytes, we used the specific NLRP3 inhibitor MCC950 to block Ca²⁺ induced IL-1 β release (supplementary figure S1b of the revised manuscript). Again, the inhibitor lead to a reduction of at least 80 % in IL-1 β , which – in view of the likely incomplete effects of the NLRP3 knockdown and the pharmacological inhibitor - indicates that the vast majority or even all of the CaSR-dependent and CaSR-independent IL-1 β maturation is mediated by NLRP3. This result was included on page 5 of the revised manuscript and in the discussion (page 15).

2. In the current study, most of calcium-induced inflammasome activation experiments were done by 16h treatment of LPS-primed monocytes. Usually, IL-1beta release from monocyte/macrophages by inflammasomes is happened in two steps, priming and activation. After priming by such as LPS, the activation step occurs within relatively short period (1-3h) except crystal/particulate-induced NLRP3 activation, which is needed 6-16h (Martinon et al., Gout-associated uric acid crystals activate the NALP3 inflammasome. 2006, Nature, 440:237-241). Thus, it is possible that calcium-CPP-mediated NLRP3 activation is not only CaSR-mediated but also crystal/particulate-induced by CPP. Actually, authors showed that calcium-induced CPP uptake was happened within 30 min (Fig. 2k). Since engulf of CPP is the important step for CaSR-mediated NLRP3 activation, the NLRP3 inflammasome is expected to be activated within few hours. Thus, author should provide data that can exclude this possibility by showing early time points and also by including additional data for both MSU and ATP

stimulations as controls in several data, particularly Fig.3b (need MSU); Fig.3c (need ATP); Fig.3f (need MSU); Fig.3g (need both ATP and MSU); Fig.4a (need both ATP and MSU); Fig. 4i (need MSU).

Response: The kinetics of NLRP3 inflammasome activation in human monocytes following stimulation with increased concentration of extracellular calcium ions has been investigated in detail previously and has been published in a 2012 report from our group. We agree with the reviewer, however, that crystal/particulate-induced NLRP3 activation requires - for unknown reasons - a considerably longer incubation period. As suggested by the reviewer, we performed additional experiments, in which calciprotein particles were used in low phosphate media for NLRP3 inflammasome activation and in which IL-1 β release was determined at earlier time points. The results of those kinetics are now shown in figure 3b. Unexpectedly, stimulation with calciprotein particles in the presence of increased concentrations of calcium ions in low phosphate media results in IL-1 β release already after 3 hours of incubation. The mechanism behind this early inflammasome activation requires further investigation.

In addition, we have repeated a number of experiments with MSU and ATP included as controls, as suggested by the reviewer. Specifically, new data with controls are shown in the following figures of the revised manuscript: 3c, 3d, 3g, and 3h. In addition, the MSU control requested by the reviewer for figure 4i (which is now figure 5 c of the revised manuscript) is included in figure 5h of the revised manuscript.

The controls suggested by the reviewer for the [Ca²⁺]_{ex}-induced IL-1 β response of RA patients (figure 4a) could be performed for MSU only due to the limited cell numbers that can be obtained from RA patients (limited to 50 ml of blood), the results are now given in a separate figure (figure S3a and S3b of the revised manuscript).

3. *In Fig 5, for the data from CAIA model and CIA model, it is suggested to include data for treatment with CaSR antagonists, which are expected to attenuate RA symptoms.*

Response: Unfortunately, due to the consequences of COVID-19, no animal experiments beyond *in vitro*/ *ex vivo* experiments with cells from sacrificed mice are possible anymore, so that further preclinical testing is currently and for the foreseeable future not possible. In addition, due to the necessary drastic reduction of the number of animals kept for line maintenance in the animal facility, extensive and time consuming breeding would be required first before animal arthritis experiments could be restarted, even after restriction will be lifted.

Reviewer #2:

Major points

It is particularly appealing that CaSR is identified as a potential target preventing macropinocytosis of calcium containing particles and NLRP3 activation. Other particles, such as MSU, also trigger the inflammasome, and deletion of NLRP3 or downstream pathway members ASC or GSDMD can temporarily restrict IL-1b production and cell death in vitro, however alternate mechanism exist to eventually trigger cell death (Rashidi et al JI 2019). Ultimately, this means that in vivo, targeting NLRP3/ASC/GSDMD had little physiological effect on MSU induced inflammation. Therefore, CaSR could potentially prevent calcium particle uptake, and all subsequent forms of downstream cell death. The authors could potentially perform some experiments to confirm this, and compare against other NLRP3 activators.

Response: We thank the reviewer for those comments, and in particular for highlighting the link between CaSR signaling and pyroptotic IL-1 β release in particle-induced NLRP3 inflammasome activation. We agree with the reviewer, that *in vivo*, the treatment outcome in particle induced inflammation depends not only on inhibition of IL-

1 β release but also on inhibition of cell death following particle engulfment. The publication by Rashidi et al. (JI 2019) showed, that MSU crystal-induced cell death is not influenced by GSDM or MLKL deficiency, Cathepsin inhibition or Caspase-1 inhibition, which makes all those agents probably not suitable for clinical therapeutic application.

As suggested by the reviewer, we performed experiments investigating the effects of inhibiting different steps of inflammasome activation, and in particular of inhibiting CaSR mediated CPP uptake, on cell death by using the NLRP3 inhibitor MCC950 and a Caspase 1 inhibitor in comparison to the CaSR inhibitor Calhex. The comparison of MSU induced vs. Ca²⁺ induced cell death showed, that inhibition of Caspase-1 as the most downstream possibility to inhibit inflammation did not reduce MSU induced cell death, while – in contrast to the result from Rashidi et al. - the specific NLRP3 inhibitor MCC950 was able to inhibit cell death. Most importantly, we found, that inhibition of both CaSR signaling and of Cathepsin activity lead to a significant reduction of Ca²⁺ induced, but not of MSU induced, cell death (figure 5h and page 14 of the revised manuscript). This indicates, that calcilytic therapy might indeed represent an option to inhibit Ca²⁺ induced inflammation and cell death in RA, which is far enough upstream of the NLRP3 inflammasome to be beneficial *in vivo*.

Reviewer #3:

The authors should explicitly comment that they evidently have both CaSR-dependent and CaSR-independent IL-1beta release in their system. The knockout of CaSR reduced, but did not abrogate, the effects they report.

Response: We thank the reviewer for this comment, which is in line with the first comment from reviewer 1. To further investigate the calcium-induced, CaSR-independent IL-1 β response and confirm its magnitude, CaSR deficient peripheral blood monocytes from ^{Lyz2tm1(cre)lfo}/CaSR ^{Δ flox/ Δ flox} mice were used in additional experiments where [Ca²⁺]_{ex}-stimulation was compared to the NLRP3 inflammasome activator MSU (as suggested by reviewer 1, results in figure 1b of the revised manuscript). The results confirmed, that approximately 50% of the calcium-induced IL-1 β response is CaSR-independent. This has been stated in more detail in the discussion on page 16 of the revised manuscript.

In addition, we performed experiments with the NLRP3 deficient THP-1 cells and the specific NLRP3 inhibitor MCC950 and found, that the CaSR-independent part of calcium-induced IL-1 β response was also NLRP3 dependent (Supplementary figure S1a,b).

As suggested by the reviewer, we explicitly stated in the discussion (page 15 of the revised manuscript), that both CaSR-dependent and CaSR-independent IL-1 β release occurs after stimulation of monocytes with [Ca²⁺]_{ex}.

CPPs were taken up by monocytes and some IL-1beta was produced under near-physiological conditions (e.g. 1 - 2 mM Ca²⁺ext), in some of the data shown. Where is the threshold for pathological inflammation?

Response: We believe, that pathological inflammation is triggered by the “solubility product” of extracellular Ca²⁺ and CPP concentration. Unfortunately, neither Ca²⁺ nor CPP concentrations in the interstitium of tissues can easily be determined. In vitro, we have seen significant IL-1 β release triggered by high CPP load at Ca²⁺ concentration of 1.1 mM in the extra-cellular fluid (corresponding to 1.5 mM added [Ca²⁺]_{ex}). 1.1mM is also the Ca²⁺ concentration determined in RA synovial fluid, and the concentration, which triggers an increased IL-1 β release of RA monocytes in the presence of pre-formed CPPs (page 12 of the revised manuscript). This has been explained in more detail in the discussion on page 15 of the revised manuscript.

A few minor points that need attention:

1. *There are a few typographical errors. For example, page 4, line 24. 'adenosin' should be 'adenosine'.*

Response: Errors have been corrected.

2. *Legend to Figure 1. '[Ca²⁺]' needs a superscript '2'.*

Response: Has been corrected.

3. *Figure 1a. Given the size of the error bars on the graph, the response to ATP doesn't look to be that significant compared to the zero calcium control. Please confirm there was a significant response to ATP. Why are the responses to ATP so different in Figures 1 and 3?*

Response: The scale of the y-axis in figure 1a was chosen to accommodate the calcium induced responses which are considerably higher than the response to ATP. Nevertheless, ATP responses were 0.3 ng/ml for ATP compared to 0.1 for LPS with a significant difference.

The difference in the magnitude of ATP response between figure 1 and figure 3 results from the fact, that figure 1a represents THP-1 cells, which were transfected with Cas9 enzyme and sgRNA and had to be cultured through several passages due to the requirements of the method (see point 10). In figure 3, in contrast, the ATP response shown is for freshly separated monocytes, which react much stronger.

4. *Page 6, line 19. '... over-representation of calcium...' doesn't really make sense ('over-representation' suggests error in measurement or presentation of data). I presume the authors mean an excess of calcium over phosphorus after 8 hours?*

Response: The wording has been corrected as suggested by the reviewer.

5. *The labelling of the y-axes in Figure 2 is non-conventional and misleading. 'Calcein uptake in %' should be replaced with 'Cells with calcein fluorescence (%)', as that is what the data show. Please try to use a similar labelling format (Figure 2i) has a slightly different y-axis label style.*

Response: The labelling has been corrected as suggested by the reviewer. Due to space restraints, we labeled the y-Axes as Calcein⁺ monocytes (%).

6. *Did the calcein enter the cells by macropinocytosis from free solution, or did the calcein bind to CPPs that were then taken into cells? Either way, the increased micropinocytosis due to Ca²⁺ext is clear, but I am wondering if the calcein uptake is proportional to CPP uptake in the assay shown.*

Response: In figure 2b-f, where y-axes are labelled 'Calcein uptake in %', experiments with calcein in free solution are shown. In figure 2i-k and in figure 4f, uptake of pre-formed and with calcein pre-stained calciprotein particles is shown. In both settings, the uptake mechanism is macropinocytosis, but since the fluorescence detection can't differentiate CPP-bound calcein from calcein taken up in free fluid, we can not comment on whether the calcein uptake is proportional to CPP uptake.

7. *Page 8, line 8. '... und...' should be '... and...'.*

Response: The error has been corrected.

8. *Figure 2i. The calcein-labelled particles with the monocytes appear to be much larger than the 60 – 100 nm particles observed in EM (even accounting for light scattering). Please explain the variations in size of the intracellular particles.*

Response: Missing scale bars in the electron microscopy images have been added in some of the images and show now, that the size of particles in the two EM techniques is comparable.

To answer the question of the reviewer regarding the variations in sizing the particles with different methods: Intracellular particles detected by fluorescence microscopy (Image stream), microscope or FACS cannot represent the actual particle size, since the magnification of light microscopy is not sufficient. Instead, the size of the calcein-positive spots is determined by the fluorescence signal intensity, and likely also represents clusters of particles in macropinosomes, phagosomes or phagolysosome. This has been explained in more detail on page 8 of the revised manuscript.

9. Please explain what the DMR measurements are indicative of. An inexperienced reader needs to understand what is being measured.

Response: A more informative explanation of the method has been inserted on page 9 of the revised manuscript.

10. The variation in amount of IL-1beta measured in different experiments is surprising. There is a 10-fold difference in the y-axis scales from different experiments (even with strong stimulation conditions). What's the basis of the variation?

Response: The differences referred to by the reviewer exist mainly between calcium-induced IL-1 β induction in figure 1,3 and figure 4. The explanation for these discrepancies comes from the fact that different cells react differently to calcium stimulation. Freshly separated peripheral blood monocytes usually give the strongest IL-1 β response after calcium stimulation. Mouse monocytes, in contrast, respond with comparatively lower (if determined by ELISA) IL-1 β concentrations (figure 1b) but produce more IL-1 α for unknown reasons. The monocytic cell line THP-1, and in particular genetically manipulated cells from this cell line, typically respond much more subdued to stimuli compared to primary monocytes, and sometimes produce only a fraction of the IL-1 β response observed in primary cells (figure 1a compared to figure 1c).

Reviewer #4:

-The EDX spectra presented should have all elements identified, not only those that are being discussed (even if due to the holder/microscope, sample preparation and support grids this can be – and in part already is – explained). This specifically relates to Figure 1J and 1M, and supplementary Figure 2B.

Response: We added all peak assignments to the respective EDX spectra in Figure 1j, 1m and 2b.

-Does the EDX spectrum in Figure 1J relate to that in Figure 1I, if so this should be stated.

Response: The EDX spectrum in Figure 1j does indeed relate to figure 1i, and this has now been stated explicitly in the figure legend in the revised manuscript. The EDX spectrum is averaged over the total image (including particles and background).

-Figure 1K is described in the caption as a TEM image, however the contrast suggests this is a HAADF STEM image and as such should be corrected.

Response: The reviewer correctly identified the image as a HAADF STEM image – the error was corrected in the figure legend of the revised manuscript.

-The figure caption for Figure 1M is confusing – what is the '(d + e)' referring to?

Response: The misleading caption in the figure legend has been corrected.

-In the methods section on 'Particle size analysis' the statement 'Initially a comparative study was performed to check if imaging artifacts due to the drying process were observed.'. Is the sizing study here for primary particle size measurements? If so, this should be stated. While primary particle size measurements are readily achieved by TEM, specialist approaches are required to prevent artefacts thereby allowing analysis of any agglomeration of particles (which I do not believe has been attempted here).

Response: Yes, this comparison is necessary, as cryo-TEM measures the particles in their solution-like state. When drying they could shrink, and hence provide unreliable information. EDX measurements however can be performed only in the dry state. Therefore, this step was included to make sure that the particles are still in the original shape after drying on the TEM-grid. It has been stated in the result section (page 6 of the revised manuscript) and in the methods section (page 54 of the revised manuscript), that the comparison between cryo-TEM and TEM was used to confirm the sizing of the particles before and after the drying process necessary to perform EDX analysis.

-In the methods section on 'Cell stimulation and preparation for TEM investigations' the sample block are presumably 1 by 1 mm (rather than mM).

Response: The reviewer is right, the error has been corrected.

Reviewer #5:

*One overarching concern is the specificity for RA and whether this is a cause or effect of inflammation. The authors argued first that RA monocytes are more sensitive to the calcium and then later state that the concentration of calcium is higher in RA and is therefore responsible for the biological effects observed. In addition, the authors show that IL-1 production is increased in CIA monocyte/macrophages compared to non-arthritic mice. Thus, the increased sensitivity to calcium could be a response to inflammation rather than a central abnormality of RA pathogenesis. **Additional controls in monocytes from patients with other forms of chronic arthritis are needed to determine whether this is an RA phenomenon or a broader response inflammation.** An additional pre-clinical models would be important to determine whether this phenomenon is specific to synovial inflammation.*

Response: We agree with the reviewer, that it is important to distinguish secondary effects due to chronic inflammation from the primary pathogenesis of RA. In order to address this question, we have recruited 2 cohorts of patients suffering from other rheumatological disorders: one group of patients with psoriatic arthritis (n=8), which is also an erosive, bone destructive disease, and a second cohort of patients with SLE (n=5). The results show that patients suffering from those diseases did not differ in their calcium induced IL-1 β response of peripheral blood monocytes from that of healthy controls.

Nevertheless, it is extremely difficult to exclude the possibility, that the observed increase of the [Ca²⁺] induced monocyte response in RA is a secondary effect of monocyte activation in this disease. Accordingly, there might be other inflammatory or autoimmune diseases, which show a similar response of their monocytes. We have pointed out this possibility in the discussion (page 17 of the revised manuscript).

While we were able to recruit several patient cohorts very quickly immediately after receiving the reviewers comments, we are now no longer able to recruit patients or age matched controls. One reason is obviously the near complete shutdown here in Germany which also extends to the lab. In addition, due to the mostly elderly patients and controls we are working with, it would not be justified to have them come in to have blood taken for research purposes alone at times of COVID-19. And finally, and most importantly, the shutdown has also

extended to the animal facility in our institution, which only maintains very few activities and animals, and animal experiments or not possible for the first seeable future. Since even after the end of the shutdown, only very few animals will be available, breeding will again take considerable time. Therefore, there is no realistic chance of performing experiments with an additional pre-clinical model as suggested by the reviewer at present time.

1. Many of the drugs used in RA are known to affect cytokine production, including prednisone and methotrexate. In most cases, they would tend to decrease cytokine production rather than increase production. Nevertheless, the authors should state what medications the RA patients were taking and determine whether cytokine release results correlated with IL-1 production?

Response: The reviewer is correct in pointing to the potentially profound effects that immunosuppressive medication can exert on the cytokine production of basically all immune cells in RA. The patient cohort presented in the original version of the manuscript represents a typical cross-sectional study with various immunosuppressant treatment regimen's including methotrexate, cytokine inhibitors, and JAK inhibitors (details now included in the section on patients and methods on page 48 of the revised manuscript), and also included a small number of patients who had not been treated with immunosuppressives prior to study enrollment.

For the revised version of the manuscript, a study cohort of patients without immunosuppressive drugs was analyzed (n=12) including 7 newly recruited patients with recent-onset rheumatoid arthritis, who had not received any treatment for RA before. The other 5 patients in this cohort had established, longstanding disease, and had not been under immunosuppression for various reasons. This patient cohort of patients without immunosuppression (n=12) has been compared to all patients who were treated with immunosuppressed drugs when they were enrolled in the study (n=39).

The results shown in figure 4a-c of the revised manuscript indicate, that the calcium induced IL-1 β response of peripheral blood monocytes is particularly high in patients not receiving immunosuppression. As expected, the clinical analysis of those 12 patients also showed a significantly higher disease activity as determined by the DAS28 compared to the rest of the study population. This indicates that the analysis of clinical associations of calcium induced IL-1 β production must be interpreted in view of the fact, that this IL-1 β secretion is very sensitive to immunosuppression by commonly used antirheumatic drugs.

2. Figure 4e shows bell shaped dose response, which is always a concern in biological models. Because the authors are attempting to mimic the effects of SF, they should directly assess the effect of SF on monocytes and whether calcium levels in SF correlate with cytokine release.

Response: We thank the reviewer for his comment. We believe that the bell shaped dose response curve is the result of a certain toxicity of increasing calcium phosphate concentrations beyond the physiological level. The suggestion of the reviewer to directly assess the effects of synovial fluid on monocytes poses a significant problem, however, since - depending of the clinical activity of the joint – the viscosity is often too high to perform reasonable cell culture. More importantly, however, in view of the difference in Ca²⁺ concentrations between RA and OA synovial fluid of 0.2 mM, and in view of the concentrations curves for monocyte stimulation with Ca²⁺ presented in the paper and the biological variations to be expected, we do not see, that such an experiment would produce meaningful results.

3. Figure 5c—Quantification and statistical significance is needed

Response: As indicated in the figure legend of figure 4c, only synovial membrane samples from two patients with osteoarthritis were available to study, which precludes meaningful statistical analysis of a quantification of expression. In the current situation of the German health system responding to COVID-19, no elective operations

or procedures on patients with osteoarthritis are performed, so that no samples can be acquired from our corporation partners.

The major message conveyed by figure 5c (6c of the revised manuscript) is, that strong expression of calcium sensor receptor is detectable in RA synovial membrane, which was analyzed in samples from 6 RA patients. Immunohistochemistry showed fairly uniform expression in all samples, one of which is given as a representative example. To demonstrate this strong expression, we would like to keep the image from the representative samples without quantifying the difference between RA and osteoarthritis.

4. Figure 5d—The differences are statistically significant, but quite small. The authors evaluate the model more completely, including effects on histology, bone damage, and synovial cytokine expression to determine whether the mechanism is related to the in vitro effects of calcium on monocytes.

Response: As already pointed out in the response to reviewer 1 (and reviewer 5, see above), no animal experiments are currently possible in our animal facility. Unfortunately, a confirmative pre-clinical experiment as suggested by the reviewer, or a more detailed histological analysis of this experiment is not possible.

5. Figure 5f—The authors state that calcium and IL-1 are “closely correlated”. The results are statistically significant, but are not particularly striking in terms of the R value.

Response: The reviewer likely refers to figure 5h of the original manuscript (figure 6h of the revised manuscript). The wording “closely” has been removed (page 15 of the revised manuscript).

6. Figure 5i—Insufficient information is provided on the protocol for CIA or the time points where cells were harvested.

Response: The protocol used for CIA is given in more detail in the material and methods section (page 50-51 of the revised manuscript). The specific time points of cell harvest are now given in the legend of figure 6e and 6g.

7. Figure 5j—Not necessary and could be deleted

Response: We believe, that due to the complex mechanism of pathological inflammasome activation proposed here, a summarizing figure is important for reference while reading the manuscript. We believe this to be an editorial decision, however.

8. Discussion—Although there is a 2006 paper suggesting that hypercalcemia is correlated with RA severity, abnormalities of calcium levels are not consistently recognized as a key feature of RA.

Response: The passage in the discussion (page 17 of the revised manuscript) has been altered according to the suggestion of the reviewer.

REVIEWER COMMENTS

Reviewer #1 (Remarks to the Author):

The authors have addressed most of my comments except one due to an inability by the COVID-19. There are still couple of minor points about appropriate control experiments as commented below.

For the comment #1, authors provided additional data from NLRP3-deficient THP-1 cells in supplementary figure S1 and stated appropriately in the Discussion for the possibility of an additional mechanism that also contribute to calcium-induced IL-1 β production.

For the results of ATP or MSU treatment in Figure 1a,b, which showed much lower IL-1 β than calcium treatment, authors stated that it is unexplained. However, comparing the same treatments in other figures of current manuscript, Figure 3c, g, h, and Figure 5c, the pattern is not consistent, and ATP treatments showed much higher IL-1 β releases. According to figure legends, 'IL-1 β was detected in supernatants after 16h'. But, usually ATP treatment to LPS-primed cells is done within 0.5-1h for NLRP3 inflammasome activation. Because ATP and MSU are used not only for positive controls but also to discriminate the specific mechanism for NLRP3 inflammasome activation by CaSR, it is very important to include data from proper experiments with a consistent result. Thus, authors' proper explanation is required.

In the same context, authors did not provide appropriate control experiments in Figure 3h. Treatments with only ATP or MSU are not reasonable controls in the figure. Author should show ATP or MSU with/without BaSO₄ particles as shown calcium treatments.

For the comments about including data of CaSR antagonists treatment in CAIA model and CIA model, it is admitted that authors were unable to conduct the experiments due to the situation of COVID-19.

Reviewer #2 (Remarks to the Author):

My comments have been addressed

Reviewer #4 (Remarks to the Author):

I was the initial reviewer 4 and all my comments have been appropriately addressed.

Reviewer #5 (Remarks to the Author):

1. The authors did a great job getting non-RA disease controls in difficult circumstances due to COVID19. However, it is not clear that their conclusion of RA-specificity is justified. The PsA and SLE groups are underpowered compared with HC and RA, so a type II error is very possible. A better way to analyze the cross-disease data is to compare 1.7 mM of Ca⁺⁺ for RA to PsA and SLE for each of the analytes to show whether RA is significantly more responsive than PsA or SLE. If not, then the data could still be shown but the conclusion that it is specific should be deleted or tempered by saying that there is a trend but that the study was underpowered to answer the question of specificity.

2. In terms of specificity, the fact that the same abnormality is observed in CIA strongly suggests that it is secondary due intense inflammation. The fact that Ca⁺⁺ concentrations are higher in RA than OA SF also suggests that there the effects are at least partially secondary. The observations are still important, but comments claiming disease specificity or that this is a primary effect should

be modified accordingly.

3. For the CIA model, it is unfortunate but understandable that COVID19 has prevented collecting additional information and providing mechanistic support for the conclusions.

4. The schema explaining the hypothesis should be either deleted or moved to a supplementary figure. The two left hand panels merely show what has been illustrated in many reviews.

We thank the reviewers for their constructive criticism. We would like to respond to their comments as follows:

Reviewer 1:

For the results of ATP or MSU treatment in Figure 1a,b, which showed much lower IL-1 β than calcium treatment, authors stated that it is unexplained. However, comparing the same treatments in other figures of current manuscript, Figure 3c, g, h, and Figure 5c, the pattern is not consistent, and ATP treatments showed much higher IL-1 β releases.

We agree with the reviewer, that the response to ATP and MSU in figure 1a is low. In our opinion, this is most likely due to the fact, that the THP1 cells were transfected with Cas9 and sgRNA, and were afterwards expanded from single cells over a prolonged period of time, and therefore do not react as strongly as “normal” THP1 cells to ATP and MSU. Nevertheless, we would like to point out, that there was still a significant increase of IL-1 β release triggered by ATP and MSU compared to LPS stimulation alone (although the absolute levels of IL-1 β are low). The graphical presentation in figure 1a was modified to reflect that fact, and the level of significance for the ATP and MSU controls in comparison to LPS was included in the graph 1a.

Regarding the results from the CaSR knockout mice shown figure 1b, we were not able to do an ATP stimulation control in the previous experiments for the revision due to cell number restraints. Following the suggestion of the reviewer, we now repeated the ATP control experiments with CaSR deficient mouse monocytes separately, using the last 10 wt vs. 10 ko mice available. We were able to document, that the ATP response of mouse monocytes did not differ between wt and ko mice (see supplementary figure S1a of the revised manuscript).

Regarding the pattern of responses to ATP in Figures 1, 3 and 5, the differences can in part be explained by the varying cell types. As mentioned before, the genetically manipulated THP1 cells respond very poorly, while freshly separated human monocytes (figure 3 and 5) produce much more IL-1 β . The results from the IL-1 β response of mouse monocytes are obtained with a different ELISA specific for murine IL-1 β , and mouse monocytes and macrophages always react with lower IL-1 β but higher IL-1 α production compared to human cells.

According to figure legends, ‘IL-1 β was detected in supernatants after 16h’. But, usually ATP treatment to LPS-primed cells is done within 0.5-1h for NLRP3 inflammasome activation. Because ATP and MSU are used not only for positive controls but also to discriminate the specific mechanism for NLRP3 inflammasome activation by CaSR, it is very important to include data from proper experiments with a consistent result. Thus, authors’ proper explanation is required.

As suggested by the reviewer, we aimed to discriminate between different mechanisms for NLRP3 inflammasome activation by comparing stimulation of LPS pre-primed monocytes with calcium and controls over 60 minutes to simultaneous stimulation with LPS and another NLRP3 activator over 16 hours. As suggested by the reviewer, we used a protocol, in which monocytes were primed for 6 hours with LPS first, and then stimulated with ATP for 30 minutes only.

The results shown in supplementary figure 1d of the revised manuscript confirm, that the highest response to ATP stimulation can be elicited by priming monocytes with LPS for six hours, followed by stimulation with ATP for 30 minutes. This is clearly different from stimulation with [Ca²⁺]_{ex}, which did not induce an IL-1 β response after 30 min in monocytes that had been primed with LPS for 6h. [Ca²⁺]_{ex} stimulation induced the highest IL-1 β response after stimulation over 16 hours with both LPS and [Ca²⁺]_{ex} simultaneously. We show this aspect, which was highlighted by the reviewer, in supplementary figure 1d of the revised manuscript.

We would also like to point out, however, that the original reason for including an ATP control stimulated over 16 h was to have a control comparable to the [Ca²⁺]_{ex} stimulation. We have already shown previously, that LPS priming of monocytes for up to 4 hours before stimulation with [Ca²⁺]_{ex} does not increase the [Ca²⁺]_{ex}-induced IL-1 β response (Rossol et al, Nat Comm 2012).

In the same context, authors did not provide appropriate control experiments in Figure 3h. Treatments with only ATP or MSU are not reasonable controls in the figure. Author should show ATP or MSU with/without BaSO₄ particles as shown calcium treatments.

As suggested by the reviewer, we repeated the barium sulfate particle experiment with simultaneous addition of either MSU or ATP. The results shown in supplementary figure 2f and 2g of the revised

manuscript indicate, that simultaneous addition of barium sulfate particles does not influence the monocyte response to MSU crystals, while there was a non-significant trend towards inhibition of the ATP response.

Again, we would like to clarify that the original goal of the experiment was to investigate, whether similarly sized particles with a composition comparable to calciprotein particles can induce the same pro-inflammatory response in monocytes as calciprotein particles, or more precisely whether calcium-induced uptake of albumin-containing barium sulfate particles triggers an IL-1 β response in monocytes comparable to the response of calcium-induced uptake of calciprotein particles. This was clearly not the case at three different particle concentrations. In conclusion, uptake of protein-containing mineral-nanoparticles does not universally induce inflammasome activation in monocytes.

Reviewer 5:

1. The authors did a great job getting non-RA disease controls in difficult circumstances due to COVID19. However, it is not clear that their conclusion of RA-specificity is justified. The PsA and SLE groups are underpowered compared with HC and RA, so a type II error is very possible. A better way to analyze the cross-disease data is to compare 1.7 mM of Ca⁺⁺ for RA to PsA and SLE for each of the analytes to show whether RA is significantly more responsive than PsA or SLE. If not, then the data could still be shown but the conclusion that it is specific should be deleted or tempered by saying that there is a trend but that the study was underpowered to answer the question of specificity.

As suggested by the reviewer, we compared the IL-1 β response (and the other analytes) induced by 1.7 mM of calcium in the three diseases directly.

We found indeed, that the newly recruited group of untreated RA patients produced significantly higher IL-1 β concentrations when compared to the psoriatic arthritis patients. We will include these results in the manuscript (page 13 of the revised manuscript).

2. In terms of specificity, the fact that the same abnormality is observed in CIA strongly suggests that it is secondary due intense inflammation. The fact that Ca⁺⁺ concentrations are higher in RA than OA SF also suggests that there the effects are at least partially secondary. The observations are still important, but comments claiming disease specificity or that this is a primary effect should be modified accordingly.

Again, we agree with the reviewer, that the observed effect is not necessarily specific for RA and might be triggered by chronic inflammation, although we would like to point out, that the comparison of calcium concentrations in synovial fluid was between RA and psoriatic arthritis and spondylitis ankylosans, but not osteoarthritis. The passages in the discussion which could imply that the described mechanism is specific for RA and cannot play a role in other inflammatory diseases were already toned down during the first round of revisions. Following the suggestion of the reviewer, we have now also modified the last paragraph of the abstract, in order to make it more clear that the reported observations don't have to be specific for RA.

REVIEWERS' COMMENTS:

Reviewer #1 (Remarks to the Author):

Authors have addressed the remaining comments appropriately.